# APPROACHING RATE-DISTORTION LIMITS IN NEURAL COMPRESSION WITH LATTICE TRANSFORM CODING

**Eric Lei, Hamed Hassani & Shirin Saeedi Bidokhti**
Department of Electrical and Systems Engineering, University of Pennsylvania
`{elei, hassani, saeedi}@seas.upenn.edu`

## ABSTRACT

Neural compression has brought tremendous progress in designing lossy compressors with good rate-distortion (RD) performance at low complexity. Thus far, neural compression design involves transforming the source to a latent vector, which is then rounded to integers and entropy coded. While this approach has been shown to be optimal on a few specific sources, we show that it can be highly sub-optimal on synthetic sources whose intrinsic dimensionality is greater than one. With integer rounding in the latent space, the quantization regions induced by neural transformations, remain square-like and fail to match those of optimal vector quantization. We demonstrate that this phenomenon is due to the choice of scalar quantization in the latent space, and not the transform design. By employing lattice quantization instead, we propose Lattice Transform Coding (LTC) and show that it approximately recovers optimal vector quantization at reasonable complexity. On real-world sources, LTC improves upon standard neural compressors. LTC also provides a framework that can integrate structurally (near) optimal information-theoretic designs into lossy compression; examples include block coding, which yields coding gain over optimal one-shot coding and approaches the asymptotically-achievable rate-distortion function, as well as nested lattice quantization for low complexity fixed-rate coding.

## 1 INTRODUCTION

Neural compression, especially nonlinear transform coding (NTC), has made significant progress in advancing practical data compression performance on real-world data at low complexity (Ballé et al., 2021; Yang et al., 2022). In this work, we seek to understand when and how neural compressors may be optimal in terms of the rate-distortion trade-off. Obtaining such an understanding can provide insights on if and how neural compressors can be further improved, as well as why they perform well.

To investigate this, it is important to understand the underlying principles behind NTC. Shown in Fig. 1, NTC operates on a source realization $\boldsymbol{x} \sim P_{\boldsymbol{x}}$ by first transforming it to a latent vector $\boldsymbol{y}$ via an analysis transform $g_a$. The latent vector is then scalar quantized to $\hat{\boldsymbol{y}}$ via $Q$, which is typically implemented via entry-wise rounding to integers. Then, a synthesis transform $g_s$ transforms $\hat{\boldsymbol{y}}$ into the reconstruction $\hat{\boldsymbol{x}}$. Additionally, an entropy model $p_{\hat{\boldsymbol{y}}}$ estimates likelihoods of the quantized latent vectors $\hat{\boldsymbol{y}}$, providing a coding rate estimate via its entropy. Together, $g_a$, $g_s$, and $p_{\hat{\boldsymbol{y}}}$ are learned to minimize a rate-distortion loss,

$$\min_{g_a, g_s, p_{\hat{\boldsymbol{y}}}} \mathbb{E}_{\boldsymbol{x}}\left[-\log p_{\hat{\boldsymbol{y}}}(\hat{\boldsymbol{y}})\right] + \lambda \cdot \mathbb{E}_{\boldsymbol{x}}[\mathsf{d}(\boldsymbol{x}, \hat{\boldsymbol{x}})], \tag{1}$$

where $\boldsymbol{y} = g_a(\boldsymbol{x})$, $\hat{\boldsymbol{y}} = Q(\boldsymbol{y})$, $\hat{\boldsymbol{x}} = g_s(\hat{\boldsymbol{y}})$, and $\lambda > 0$ controls the rate-distortion tradeoff. NTC is considered a one-shot coding scheme of $\boldsymbol{x}$, because it compresses one source realization at a time.

Optimal compression of $\boldsymbol{x}$ can be performed by vector quantization (VQ) Gersho & Gray (2012) albeit at high complexity which grows exponentially in rate and dimension. NTC, on the other hand, has low complexity. We are interested in understanding if NTC can recover VQ, and/or how its performance can be improved. If so, this would provide a low-complexity, source agnostic scheme capable of achieving the fundamental limits of compression.

Wagner & Ballé (2021); Bhadane et al. (2022) demonstrate that on certain synthetic sources, NTC successfully achieves VQ performance. These sources are nominally high-dimensional, but have

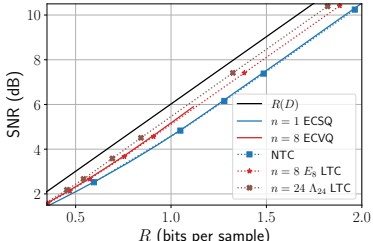 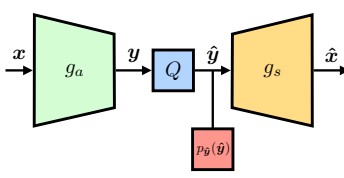

Figure 1: *Left*: Nonlinear Transform Coding (NTC) recovers scalar quantization (ECSQ) on Gaussian i.i.d. sequences of any dimension $n$. Lattice Transform Coding (LTC) recovers optimal vector quantization (VQ) in each dimension, and approachces the rate-distortion function $R(D)$ without needing an exponential codebook search. *Right*: NTC architecture. LTC replaces scalar quantizer $Q$ with a lattice quantizer $Q_\Lambda$.

a low-dimensional representation, which we refer to as the *latent source*. The authors show that $g_a, g_s$ can successfully extract the latent source, optimally quantize it, and map back to the original space, providing an optimal compression scheme as a whole. However, most of the sources they analyze have a one-dimensional uniform latent source $U \sim \mathrm{Unif}([a, b])$. While these are sufficient to analyze the role of $g_a$ and $g_s$, it does not provide insights on the role of the quantizer $Q$. This is because a uniform source $U$ can be optimally quantized using uniform scalar quantization (SQ), which is exactly what NTC uses in the latent space. For sources with higher-dimensional latent sources, where uniform SQ is not necessarily optimal, it is unclear whether NTC can still achieve optimality. Conceptually, given the high representation power of neural networks, one may hope that the transforms $g_a$ and $g_s$, in conjunction with the basic integer rounding in the latent space, can implement a more complex quantization such as VQ.

In this paper, we first provide evidence that on sources with higher latent dimensionality, NTC performs sub-optimally; the learned transforms are insufficient to overcome the sub-optimal latent SQ. Shown in Fig. 3, NTC quantization regions remain square-like (following integer rounding in the latent space), and do not match the hexagonal regions from VQ. Our observation is also aligned with the recent analytical approach of Shevchenko et al. (2023) for two-layer VAEs, showing fundamental incapability in recovering VQ for Gaussian sources. To improve performance, we propose to use lattice quantization (LQ), which provides improved space-packing efficiency over SQ, but reduced complexity compared to VQ. We refer to this framework (i.e., LQ in the latent space) as Lattice Transform Coding (LTC). On a variety of synthetic and real-world sources, we demonstrate how LTC improves upon NTC in terms of the rate-distortion trade-off, while avoiding the high complexity of a direct codebook search under VQ. For example, on isotropic Gaussian vectors, Fig. 1 shows how LTC achieves VQ performance without a codebook search.

The proposed framework of LTC further enables the integration of structurally-optimal solutions from information theory. In particular, we show how LTC enables design of (1) lower complexity solutions using nested lattices; and (2) near-optimal solutions using block coding from information theory. Overall, our experiments support LTC as a framework to achieve rate-distortion limits while mitigating high complexity. Our contributions are as follows[1].

1) We first demonstrate the inability of current NTC architectures to optimally compress sources with latent dimensionality larger than one; we conclude this is due to the choice of SQ in the latent space, and not the transform design.

2) We propose Lattice Transform Coding (LTC), a framework that replaces the latent integer rounding of NTC with general lattice quantization. LTC requires novel solutions for training, such as efficient lattice quantization algorithms, dealing with the non-differentiable quantization operation, integration of the entropy model over lattice regions, and accurate likelihood modeling.

3) LTC additionally enables integration of a variety of information-theoretic concepts into neural compression. Namely, we demonstrate how LTC allows the use of nested lattices in the latent space which provides a scheme with low encoding complexity, as well as implementation of block coding (necessary to achieve $R(D)$) which has applications in dataset compression.

---

[1]Code can be found at https://github.com/leieric/lattice-transform-coding.

4) We show the LTC framework can near-optimally compress interesting synthetic sequences for which we have a benchmark of optimality (e.g. i.i.d. sources, the Banana source), while maintaining reasonable complexity. Our results reveal the interplay between nonlinear transforms, entropy models and the choice of LQ.

5) For real-world sources with moderate to high dimensions, we demonstrate LTC's ability to consistently improve upon the performance of NTC when the latent dimension is larger than one, and bridge up to 30% of the gap to $R(D)$.

## 2 RELATED WORK

**Rate-Distortion Theory.** Given a source $X \sim P_X$, the rate-distortion function $R(D)$ describes the fundamental limit of compressing $X$ (Cover & Thomas, 2006). Compressing the source one sample at a time is known as one-shot coding. $R(D)$, on the other hand, is achievable asymptotically in the limit of large block-lengths when the source is compressed block by block and the rate and distortion are measured per-sample. For a general vector source $\boldsymbol{x}$, its optimal one-shot coding scheme can be found via entropy-constrained vector quantization (ECVQ). ECVQ directly enumerates quantization centers (codewords) in the space of $\boldsymbol{x}$, and minimizes an entropy-distortion objective,

$$\min_{\{\boldsymbol{c}_i\}_{i=1}^{\infty}} \mathbb{E}_{\boldsymbol{x}}[-\log P(e(\boldsymbol{x}))] + \lambda \cdot \mathbb{E}_{\boldsymbol{x}}[\mathsf{d}(\boldsymbol{x}, e(\boldsymbol{x}))], \tag{2}$$

where $e(\boldsymbol{x}) := \arg\min_{\{\boldsymbol{c}_i\}_{i=1}^{\infty}} \mathsf{d}(\boldsymbol{x}, \boldsymbol{c}_i)$. Since the solution is unknown in closed form, ECVQ is solved numerically by a generalized Lloyd's algorithm (Chou et al., 1989). Although ECVQ is optimal, an issue that precludes its wider use is its high complexity; the number of codebook vectors required is exponential in the rate and dimension. This complexity issue is one reason why NTC is appealing, since scalar quantization in the transformed space is used, whose complexity is linear in the dimension of $\boldsymbol{y}$. However, it is unknown whether NTC achieves optimality; this is an important question, since if NTC was optimal it would immediately offer a low-complexity solution that achieves ECVQ performance. Our work reveals that NTC is not optimal; however, LQ offers an optimal low-complexity solution for NTC.

**Nonlinear Transform Coding.** The principles of nonlinear transform coding (NTC) are described in Ballé et al. (2021). Although NTC is based on VAEs with SQ in the latent space, several works have attempted to apply VQ in the latent space instead. Notable works include VQVAE (van den Oord et al., 2017), which performs a direct codebook search in the latent space. PQVAE (El-Nouby et al., 2023) uses a product quantization scheme to improve scalability. Additionally, NVTC (Feng et al., 2023) applies ECVQ in the latent space, which also performs a direct codebook search. We show that LTC is able to recover the performance of ECVQ while *avoiding* a direct codebook search in the latent space, which is infeasible for large rates and dimensions.

**Lattice Quantization.** A lattice quantizer $Q_\Lambda$ (Conway & Sloane, 1999) consists of a countably infinite set of codebook vectors $\Lambda$ given by $\boldsymbol{u}\boldsymbol{G}$, where $\boldsymbol{G} \in \mathbb{R}^{n \times n}$ is the generator matrix, and $\boldsymbol{u}$ are integer vectors. The Voronoi region of a lattice is $V(\boldsymbol{0}) := \{\boldsymbol{x} \in \mathbb{R}^n : Q_\Lambda(\boldsymbol{x}) = \boldsymbol{0}\}$, where $Q_\Lambda(\boldsymbol{x}) := \arg\min_{\boldsymbol{v} \in \Lambda} \|\boldsymbol{x} - \boldsymbol{v}\|_2^2$ finds the closest lattice vector to $\boldsymbol{x}$ in $\Lambda$. In lattice theory, the objective is to find lattices that minimize $L_\Lambda$, the normalized second moment (NSM), which measures the lattice's packing efficiency. Recent work describing lattices which have the lowest known NSM can be found in (Agrell & Allen, 2023; Lyu et al., 2022; Agrell et al., 2024). Another objective is to find efficient quantization (decoding) algorithms (i.e., computing $Q_\Lambda(\boldsymbol{x})$), which requires solving the closest-vector problem (CVP); this is NP-hard in general. For many of the best-known lattices in dimensions up to 24 ($n = 2$: hexagonal; $n = 4$: $D_n^*$; $n = 8$: Gosset $E_8$; $n = 16$: $\Lambda_{16}$ Barnes-Wall; and $n = 24$: $\Lambda_{24}$ Leech) efficient CVP solvers exist (Conway & Sloane, 1999).

Classically, lattices are used to implement both fixed and variable-rate quantizers (Zamir et al., 2014). For the former, nested lattices shape the used part of the lattice and create a finite codebook with index-based encoding. For the latter, the entire (infinite) lattice is used, but entropy coding is applied; this is known as entropy-constrained lattice quantization (ECLQ). Fixed-rate quantization trades lower encoding complexity for rate-distortion performance. For both cases, in the large rate regime, LQ achieves optimality when the dimension is asymptotically large. For example, for ECLQ,

$$\lim_{D \to 0} \frac{1}{n}(H_{\text{ECLQ}}(D) - R_{\boldsymbol{x}}(D)) = \frac{1}{2}\log(2\pi e L_\Lambda), \tag{3}$$

where $R_{\boldsymbol{x}}(D)$ is the rate-distortion function of $\boldsymbol{x}$ and $H_{\text{ECLQ}}(D)$ is the rate of ECLQ at distortion $D$ (Zamir et al., 2014, Ch. 5). If $G_n$ is the minimal NSM of any lattice in $n$ dimensions, $\lim_{n\to\infty} G_n = 1/(2\pi e)$, and thus (3) converges to 0. For the integer lattice (SQ), the NSM is $1/12$, which yields a gap of $\approx 0.255$ bits per dimension. Thus, LQ offers the maximal benefit when the rate-per-dimension is relatively small. While this may seem limiting, many real-world sources operate at low rate-per-dimension. For example, in NTC-based image compression, the rate-per-latent dimension is around the same as the bits-per-pixel used, which is typically between 0 and 2. While classical transform coding may not have benefited much from LQ since the transform domain of images (e.g, DCT) is typically sparse, this is not true in NTC where the rate is spread more evenly across all dimensions (Cheng et al., 2020; Bhadane et al., 2021) resulting in a lower bit-per-dimension for the quantizer.

**Companding.** Companding defines a function $f : \mathbb{R}^n \to \mathbb{R}^n$, known as the compressor, which is applied to the source $\boldsymbol{x} \in \mathbb{R}^n$. $f(\boldsymbol{x})$ is then uniform *lattice* quantized, and passed to $f^{-1}$. Clearly, NTC can be thought of as an instantiation of a compander where $g_a, g_s$ serve the role of $f, f^{-1}$. Companding has typically been analyzed in the (asymptotically) large rate, small distortion regime. Optimal scalar quantization can be implemented using scalar companding (Bennett, 1948). For the vector case, vector companders cannot implement optimal VQ (Gersho, 1979; Bucklew, 1981; 1983), even when the best lattices are used, except for a very small class of sources. Despite this, it is still possible optimal companding may perform *near*-optimal VQ. This is supported by (Linder et al., 1999) which has a similar asymptotic result as (3). Due to this, we seek to utilize them in our learning-based framework and learn the companding function as guided by the companding literature.

**Lattice Quantization and Neural Compression.** The first work to integrate a lattice quantizer with NTC was LVQAC (Zhang & Wu, 2023), which proposed the use of the $D_n^*$ lattice in the latent space, and showed some performance improvement over scalar quantization. However, LVQAC does not generalize to other lattices, and the $D_n^*$ is known to perform poorly compared to many other known lattices. In contrast, our approach is generalizable to high-performing lattices, and we further show how the performance of learned compressors depend on the space-packing efficiency of the lattice quantizer. LVQ-VAE (Kudo et al., 2023), a contemporary work to ours, seeks to exploit correlations among latent features by extending the hyperprior entropy model (Ballé et al., 2018) with a covariance matrix along quantized vectors. To mitigate the costly search over vector codewords, Kudo et al. (2023) uses lattice quantization in the latent space in a manner similar to our work. However, their work does not shed light on the role of lattice quantization in NTC. Moreover, their approach suffers from high complexity. In contrast, we seek to understand the precise role that quantization plays in NTC, and lattice quantization emerges as a solution to achieve optimal rate-distortion performance. We additionally show how nested lattices can help achieve optimality at much lower complexity. Finally, Cao et al. (2024) is another contemporary paper that provides a low-complexity lattice VQ solution for image compression; however, details are sparse sin their manuscript for comparison.

## 3 NTC IS SUB-OPTIMAL

Let $\boldsymbol{x} \sim P_{\boldsymbol{x}}$ be the vector source. We train an NTC model following Ballé et al. (2021) using a factorized entropy model for the distribution of the latent $\boldsymbol{y}$. We use the objective in (1), with $\mathsf{d}(\boldsymbol{x}, \hat{\boldsymbol{x}}) = \|\boldsymbol{x} - \hat{\boldsymbol{x}}\|_2^2$, and compare with (optimal) entropy-constrained vector quantization (ECVQ). We first consider the case when the $n$ entries of $\boldsymbol{x}$ are i.i.d. with marginal $S \sim \mathcal{N}(0, 1)$. Shown in Fig. 1, NTC with $n = 1$ is able to recover ECVQ with $n = 1$, also known as entropy-constrained scalar quantization (ECSQ). This is perhaps expected, since the task of the transforms is to scale the 1-D Gaussian to be optimally quantized on an integer grid in 1-D, and then scale the quantized latent to form the reconstruction. Since the source is intrinsically one-dimensional, scalar quantization appears to be sufficient in the latent space to implement optimal quantization.

For $n = 2$, however, NTC is unable to recover ECVQ, and always achieves the same performance as that of ECSQ. Shown in Fig. 2, we see that the rate-distortion performance for ECVQ at $n = 2$ is superior to that of ECSQ. However, NTC at $n = 2$ merely recovers ECSQ performance, and fails to outperform it. Visualizing the quantization regions on the 2-D Gaussian source (Fig. 3) reveals why. NTC is merely implementing a scalar quantizer of the source, perhaps with a rotation, which is equivalent to applying ECSQ to the entries of $\boldsymbol{x}$ independently. In 2-D, this induces square quantization regions in the source space. In contrast, the optimal ECVQ in $n = 2$ has hexagonal regions, not square. A similar phenomenon occurs for the Banana source (Ballé et al., 2021). It

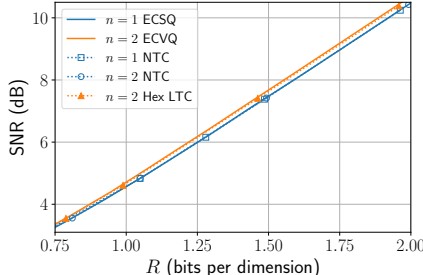

Figure 2: Rate-distortion, 2-D Gaussians.

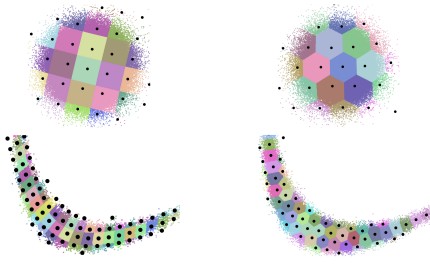

Figure 3: Quantizer regions. Left: NTC. Right: ECVQ. Top: 2-D Gaussian source. Bottom: Banana source.

is well-known that hexagonal tessellations yield superior packing efficiency compared to square tessellations (Conway & Sloane, 1999). Thus, in 2-D, for NTC to be optimal, the quantization regions (induced by latent SQ and learned transforms) in the source space need to be hexagonal.

We found that when the latent space is rounded to integers, varying the depth, width, nonlinearities, biases, and even increasing the latent dimension to a higher-dimensional space did not help; all resulted in square regions in the source space. For the i.i.d. Gaussian case, setting $n > 2$ did not help either; all models recovered the performance of ECSQ at best. This surprising observation reveals that the learned transforms $g_a$ and $g_s$ seem unable to overcome use of scalar quantization. Moreover, it confirms that when the latent dimensionality of the source is greater than one, using SQ in the latent space, which is optimal for uniform scalar sources, does not result in optimality.

Several related works in the literature appear to support this observation. Shevchenko et al. (2023) analyzes 2-layer VAEs with a form of scalar quantization in the latent space, and show that the optimal performance is given by SQ, not VQ. Results in the companding literature (Gersho, 1979; Linder et al., 1999) additionally suggest that a compressor such as NTC is (i) unable to perform exact VQ for most sources, and that (ii) the performance is determined by the choice of the lattice in the latent space, which for NTC, is equivalent to the (sub-optimal) integer lattice.

To achieve VQ performance, these observations suggest that learned transforms are insufficient in overcoming sub-optimal quantization in the latent space, and that improving said latent quantization is required to achieve optimality. As will be shown, the proposed LTC is able to recover ECVQ regions and performance, with increasing gains over NTC as dimensionality increases (Fig. 1).

## 4 LATTICE TRANSFORM CODING

We seek a compressor that can provide better performance than scalar quantization yet avoid the high complexity of vector quantization. Using a lattice quantizer $Q_\Lambda$ provides an intermediate solution that manages both complexity and space-filling efficiency. It generalizes scalar quantization, which is equivalent to the integer lattice. It is known that many lattices can provide far more space-filling efficiency compared to the integer lattice. We first discuss how to extend current NTC designs to integrate the use of lattices. We then discuss a method to reduce the entropy coding complexity of LTC by using nested lattice quantization. Finally, we discuss how LTC enables one to approach the asymptotic rate-distortion function $R(D)$ for any source.

### 4.1 FROM SCALAR TO LATTICE QUANTIZATION

The primary challenges when integrating lattice quantization with (1) include computing the quantization itself and computing integrals over the lattice cell for likelihood modelling.

**Performing lattice quantization.** The forward-pass of the lattice quantizer $Q_\Lambda$ requires the CVP algorithm of the lattice; exact algorithms we use for the lattices in this paper are described in Sec. A.1.

To allow differentiation for the backwards pass, we follow standard practices in NTC and use either the straight-through estimator (STE) or dithered quantization with uniform noise over the lattice cell. The STE approach can use (1) directly as an objective during training, but may lead to training instabilities. The uniform noise proxy has stabler training, but requires a rate-distortion proxy loss of

(1) using a connection to dithered quantization (Ballé et al., 2021). We also assume that the lattice's generator matrix $\boldsymbol{G}$ has been normalized to be unit volume, i.e. $\mathrm{Vol}(V(\boldsymbol{0})) = \sqrt{\det(\boldsymbol{G}\boldsymbol{G}^\top)} = 1$.

We use STE for the distortion term $\mathbb{E}_{\boldsymbol{x}}[\mathrm{d}(\boldsymbol{x}, g_s(Q_\Lambda(g_a(\boldsymbol{x}))))]$. Specifically, we compute the quantized output $\hat{\boldsymbol{y}} = Q_\Lambda(g_a(\boldsymbol{x}))$ during the forward pass via

$$\hat{\boldsymbol{y}} = \boldsymbol{y} + \texttt{stop\_gradient}(Q_\Lambda(\boldsymbol{y}) - \boldsymbol{y}). \tag{4}$$

The rate term, however, requires more care.

**Estimating the rate term.** For the rate term, we typically have some parameterized density model $p_{\boldsymbol{y}}$ for the continuous latent $\boldsymbol{y}$ (more details on this in the next section). This induces a PMF over the quantized latents $p_{\hat{\boldsymbol{y}}}$, given by $p_{\hat{\boldsymbol{y}}}(\hat{\boldsymbol{y}}) = \int_{V(\hat{\boldsymbol{y}})} p_{\boldsymbol{y}}(\boldsymbol{y})d\boldsymbol{y}$, where $V(\hat{\boldsymbol{y}}) := \{\boldsymbol{y} : Q_\Lambda(\boldsymbol{y}) = \hat{\boldsymbol{y}}\}$. In NTC, where $Q_\Lambda$ implements the integer lattice, $p_{\hat{\boldsymbol{y}}}(\hat{\boldsymbol{y}})$ is easy to integrate when $p_{\boldsymbol{y}}$ is assumed factorized (Ballé et al., 2018). In that case, $p_{\hat{\boldsymbol{y}}}(\hat{\boldsymbol{y}})$ can be written in terms of the CDF of $p_{\boldsymbol{y}}$, which is one reason why the factorized $p_{\boldsymbol{y}}$ is typically parameterized by its CDF in NTC. However, for lattice quantization, solving the integral in $p_{\hat{\boldsymbol{y}}}(\hat{\boldsymbol{y}})$ is not as straightforward.

As a result, we compute $p_{\hat{\boldsymbol{y}}}(\hat{\boldsymbol{y}})$ via Monte-Carlo estimate,

$$p_{\hat{\boldsymbol{y}}}(\hat{\boldsymbol{y}}) = \int_{V(\hat{\boldsymbol{y}})} \frac{p_{\boldsymbol{y}}(\boldsymbol{y})}{p_{\boldsymbol{u}}(\boldsymbol{y})} p_{\boldsymbol{u}}(\boldsymbol{y})d\boldsymbol{y} = \mathbb{E}_{\boldsymbol{u}'\sim\mathrm{Unif}(V(\boldsymbol{0}))}[p_{\boldsymbol{y}}(\hat{\boldsymbol{y}} + \boldsymbol{u}')], \tag{5}$$

where $p_{\boldsymbol{u}}$ is the uniform density over $V(\hat{\boldsymbol{y}})$, $\mathrm{Vol}(V(\hat{\boldsymbol{y}})) = 1$ since we assumed unit-volume lattices, and the last equality holds because for a lattice quantizer $Q_\Lambda(\boldsymbol{x} + \boldsymbol{v}) = Q_\Lambda(\boldsymbol{x}) + \boldsymbol{v}, \forall \boldsymbol{v} \in \Lambda$. From Conway & Sloane (1984), we can sample $\boldsymbol{u}' \sim \mathrm{Unif}(V(\boldsymbol{0}))$ by computing $\boldsymbol{u}' = \tilde{\boldsymbol{u}} - Q_\Lambda(\tilde{\boldsymbol{u}})$, where $\tilde{\boldsymbol{u}} = \boldsymbol{s}\boldsymbol{G}$, and $\boldsymbol{s} \sim \mathrm{Unif}([0, 1])$. Thus, (5) can be estimated via a sample mean. With the ability to reliably estimate $p_{\hat{\boldsymbol{y}}}(\hat{\boldsymbol{y}})$, we can also use STE for backpropagation of the rate term $\mathbb{E}_{\boldsymbol{x}}[-\log p_{\hat{\boldsymbol{y}}}(\hat{\boldsymbol{y}})]$ by using (4). Prior works have achieved benefits from using the dithering approach (Ballé et al., 2021). In this case, a random dither $\boldsymbol{u}' \sim \mathrm{Unif}(V(0))$ is applied prior to quantization, with the resulting rate $\mathbb{E}_{\boldsymbol{x},\boldsymbol{u}'}[-\log p_{\hat{\boldsymbol{y}}}(Q_\Lambda(\boldsymbol{y} - \boldsymbol{u}'))]$. This rate is equivalent in the integer lattice to $\mathbb{E}_{\boldsymbol{x},\boldsymbol{u}'}[-\log p_{\boldsymbol{y}+\boldsymbol{u}'}(\boldsymbol{y} + \boldsymbol{u}')]$ (Ballé et al., 2021). The density $p_{\boldsymbol{y}+\boldsymbol{u}'}$ is a convolved density with likelihoods given by

$$p_{\boldsymbol{y}+\boldsymbol{u}'}(\boldsymbol{y}') = \int_{V(0)} p_{\boldsymbol{y}}(\boldsymbol{y}' - \boldsymbol{u}')p_{\boldsymbol{u}'}(\boldsymbol{u}')d\boldsymbol{u} = \mathbb{E}_{\boldsymbol{u}'\sim\mathrm{Unif}(V(0))}[p_{\boldsymbol{y}}(\boldsymbol{y}' - \boldsymbol{u}')], \tag{6}$$

which can be estimated via Monte-Carlo. To apply this to LQ during training, rather than provide the STE of the quantized output $\hat{\boldsymbol{y}}$ to the Monte-Carlo likelihood estimate, we provide the noise-added continuous latent $\boldsymbol{y} + \boldsymbol{u}'$, where $\boldsymbol{u}'$ is uniformly distributed over the Voronoi region. In our experiments, we observed that STE was less stable than the dithering approach, but had better performance at low rates, as will be shown in Sec. 5.

We note that a concurrent work (Kudo et al., 2023) utilizes a similar approach to compute the rate term for the use of general lattices in image compression. Their goal was to improve coding performance by removing correlation among latent features, whereas we seek to understand the fundamental nature of quantization in nonlinear transform coding. Comparing to other related work such as LVQAC (Zhang & Wu, 2023), we show later how their use of $D_n^*$ lattice results in fundamental suboptimality.

**Entropy Models.** The rate term, $\mathbb{E}_{\boldsymbol{x}}[-\log p_{\hat{\boldsymbol{y}}}(\hat{\boldsymbol{y}})]$, which is the cross-entropy between the true PMF on the lattice vectors $P(\hat{\boldsymbol{y}}) := \mathrm{Pr}_{\boldsymbol{x}}(Q_\Lambda(g_a(\boldsymbol{x}) = \hat{\boldsymbol{y}})$ and $p_{\hat{\boldsymbol{y}}}(\hat{\boldsymbol{y}})$, is an upper bound on the true entropy of $P(\hat{\boldsymbol{y}})$, $\mathbb{E}_{\boldsymbol{x}}[-\log P(\hat{\boldsymbol{y}})]$, with equality if the learned PMF matches the true PMF exactly, $p_{\hat{\boldsymbol{y}}}(\hat{\boldsymbol{y}}) = P(\hat{\boldsymbol{y}})$. Using the cross-entropy as an estimate of the true entropy is critical due to the limitations of entropy estimation. Whereas direct entropy estimation requires a number of samples exponential in the rate, cross-entropy upper bounds are able to provide accurate estimates at large rates without requiring exponential samples (McAllester & Stratos, 2020). We cannot always expect the bound to be tight, and we will show that the choice of entropy model does play a role in whether optimality can be achieved in Sec. 5.3. Since we are now using a more general Monte-Carlo integration for the likelihoods of $\hat{\boldsymbol{y}}$, we are no longer tied to a factorized density and instead propose to parameterize the PDF $p_{\boldsymbol{y}}$ in a multivariate fashion using a normalizing flow (Kobyzev et al., 2020).

## 4.2 Toward Lower Complexity

However, as the dimension of the lattice increases, entropy coding becomes challenging due to the large alphabet size. Large alphabet entropy coding is indeed an active area of research. Another

approach that circumvents the need for entropy coding altogether is by quantization using nested lattices. To implement a fixed-rate code in the latent space using lattices, we use a nested lattice in the latent space. The idea is to use the Voronoi region of a coarse lattice $\Lambda_c$ to define a bounded region of a fine lattice $\Lambda_f$. The fixed number of codebook vectors of $\Lambda_f$ contained in the said region are the ones we wish to use to encode the source via index coding (and thus a fixed rate). This happens when the majority of the latent mass is placed in the Voronoi region of the coarse lattice. Efficient encoding/decoding is enabled by transmitting the index of the coset, yielding a complexity of $O(n)$ for an alphabet of size $O(2^n)$, compared to entropy coding which will be $O(2^n)$. To quantize the latent $\boldsymbol{y}$, we compute $\hat{\boldsymbol{y}} = \boldsymbol{y}_f - Q_{\Lambda_c}(\boldsymbol{y}_f)$, where $\boldsymbol{y}_f = Q_{\Lambda_f}(\boldsymbol{y})$. This operation selects the coset leader corresponding to $\boldsymbol{y}_f$ inside the Voronoi region of $\Lambda_c$. The rate is thus given by $\log_2 |\Lambda_1/\Lambda_2| = \Gamma^{d_y}$, where $|\Lambda_1/\Lambda_2|$ is the number of relative cosets, $d_y$ is the dimension of $\boldsymbol{y}$, and $\Gamma$ is the nesting ratio (Zamir et al., 2014).

The majority of the latent mass is placed in the Voronoi region of $\Lambda_c$ when $Q_{\Lambda_c}(\boldsymbol{y}_f) = \boldsymbol{0}$ occurs with high probability. Otherwise, we have an error event $\mathcal{E} := \{Q_{\Lambda_c}(\boldsymbol{y}_f) \neq \boldsymbol{0}\}$. Let $P_e = \Pr(\mathcal{E})$. The total distortion can be broken up into two terms (Zamir et al., 2014, Ch. 10),

$$\mathbb{E}_{\boldsymbol{x}}[\mathsf{d}(\boldsymbol{x}, \hat{\boldsymbol{x}})] = (1 - P_e) \cdot \mathbb{E}_{\boldsymbol{x}}\Big[\mathsf{d}(\boldsymbol{x}, \hat{\boldsymbol{x}})|\mathcal{E}^{\complement}\Big] + P_e \cdot \mathbb{E}_{\boldsymbol{x}}[\mathsf{d}(\boldsymbol{x}, \hat{\boldsymbol{x}})|\mathcal{E}]. \tag{7}$$

In order to train LTC with latent NLQ, we need to be able to differentiate through these terms. We use a normalizing flow density $p_{\boldsymbol{y}}$ to estimate the $P_e$ with Monte-Carlo integration,

$$\hat{P}_e := 1 - \int_{V_c(\boldsymbol{0})} p_{\boldsymbol{y}}(\boldsymbol{y}) d\boldsymbol{y} = 1 - \mathrm{Vol}(V_c(\boldsymbol{0})) \cdot \mathbb{E}_{\boldsymbol{u} \sim V_c(\boldsymbol{0})}[p_{\boldsymbol{y}}(\boldsymbol{u})], \tag{8}$$

where $V_c(\boldsymbol{0})$ is the Voronoi region of $\Lambda_c$. To enforce $p_{\boldsymbol{y}}$ to model the true distribution of $\boldsymbol{y} = g_a(\boldsymbol{x})$, we maximize its log likelihood $\mathbb{E}_{\boldsymbol{x}}[\log p_{\boldsymbol{y}}(g_a(\boldsymbol{x}))]$. Thus, our complete loss is given by

$$\min_{g_a, g_s, p_{\boldsymbol{y}}} (1 - \hat{P}_e) \cdot \mathbb{E}_{\boldsymbol{x}}\Big[\mathsf{d}(\boldsymbol{x}, \hat{\boldsymbol{x}})|\mathcal{E}^{\complement}\Big] + \hat{P}_e \cdot \mathbb{E}_{\boldsymbol{x}}[\mathsf{d}(\boldsymbol{x}, \hat{\boldsymbol{x}})|\mathcal{E}] + \mathbb{E}_{\boldsymbol{x}}[-\log p_{\boldsymbol{y}}(g_a(\boldsymbol{x}))]. \tag{9}$$

In our experiments, we found that detaching the gradient from the $\mathbb{E}_{\boldsymbol{x}}[\mathsf{d}(\boldsymbol{x}, \hat{\boldsymbol{x}})|\mathcal{E}]$ term helped to significantly improve training stability. For the lattices used, we used self-similar nested lattices: $G_c = \Gamma \cdot G_f$, where $G_c$ and $G_f$ are the coarse and fine lattices' generator matrices respectively, and $\Gamma$ is the integer-valued nesting ratio, used to vary the rate of the compressor.

## 4.3 FROM ONE-SHOT TO BLOCK CODING

LTC can be directly applied to any vector source $\boldsymbol{x} \in \mathbb{R}^d$. In this case, LTC operates as a one-shot code for compressing $\boldsymbol{x}$. In Sec 5, we discuss the improvements that LTC provides compared to NTC in the one-shot sense. Rate-distortion theory informs us that to go beyond one-shot performance and approach asymptotic optimality (given by $R_{\boldsymbol{x}}(D)$), we need to encode i.i.d. copies $\boldsymbol{x}_1, \dots, \boldsymbol{x}_n \sim p_{\boldsymbol{x}}$ simultaneously. This is highly relevant in settings such as dataset compression, when we wish not just to compress a single sample, but an entire dataset where samples can be assumed i.i.d. Note that $n$ remains the block-length, but $d$ is now the dimension of $\boldsymbol{x}$.

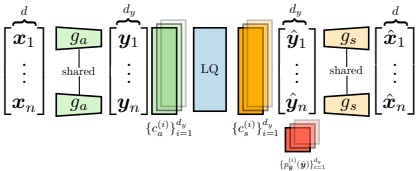

Figure 4: Block-LTC (BLTC) applies LTC in the latent space for vector i.i.d. sequences.

To realize such a compressor, we propose Block-LTC (BLTC), which applies LTC across the i.i.d. latent variables of $\boldsymbol{x}$, shown in Fig. 4. Specifically, the analysis transform $g_a : \mathbb{R}^d \to \mathbb{R}^{d_y}$ maps each vector of the block to its latent variable, $\boldsymbol{y}_i = g_a(\boldsymbol{x}_i)$, which has $d_y$ dimensions. This produces a latent i.i.d. vector sequence $\boldsymbol{y}_1, \dots, \boldsymbol{y}_n$. Then, $d_y$ separate LTC models $\{c_a^{(i)}, c_s^{(i)}, p_{\hat{\boldsymbol{y}}}^{(i)}\}_{i=1}^{d_y}$ are applied to across the block, where $c_a^{(i)}, c_s^{(i)}$ are the analysis/synthesis transforms of LTC. The $i$-th LTC model is applied to the latent slice $[\boldsymbol{y}_{1,i}, \dots, \boldsymbol{y}_{n,i}]^{\top}$, which has i.i.d. scalar entries. The resulting reconstructed latents $\hat{\boldsymbol{y}}_i$'s are passed to the synthesis transform $g_s : \mathbb{R}^{d_y} \to \mathbb{R}^d$, such that $\hat{\boldsymbol{x}}_i = g_s(\hat{\boldsymbol{y}}_i)$. We use STE to train the BLTC models. The BLTC architecture first transforms the i.i.d. vectors into i.i.d. latent vectors $\boldsymbol{y}_i$; the space-packing ability of LTC compresses the i.i.d. latent vectors.

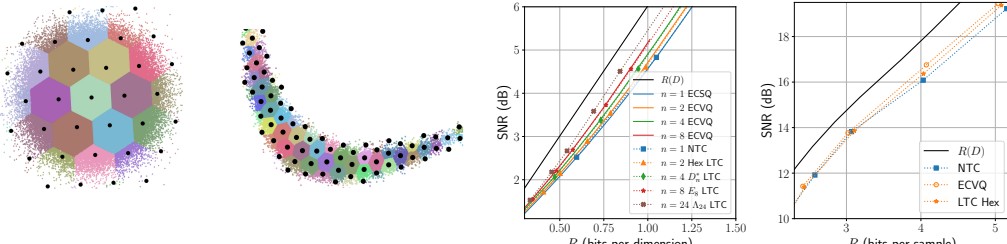

Figure 5: LTC ($A_2$ lattice) quantization regions. *Left*: 2-D Gaussians. *Right*: Banana.

Figure 6: LTC on synthetic sources. *Left*: Gaussians. *Right*: Banana.

## 5 RESULTS

We first discuss LTC on synthetic sources, before moving to real-world sources. Then, we demonstrate how LTC performs with nested lattices, and finally present results on block coding using BLTC. In addition, we provide an ablation study demonstrating the effects of various components in LTC.

**Experimental setup.** For LTC's rate term, we found varying results for dithering (6) and STE (5) depending on the source; these are specified for each experiment individually. Additional implementation details can be found in Appendix A. We choose lattices best in their dimension (Agrell & Allen, 2023): $A_2$ (hexagonal), $D_4^*$, $E_8$ and $\Lambda_{24}$ (Leech) lattices. These lattices all have efficient algorithms to solve the closest vector problem (CVP), which computes $Q_\Lambda(\boldsymbol{y})$. For the Euclidean vector sources, we choose the transforms $g_a$ and $g_s$ to be MLPs with two hidden layers, a hidden dimension of 100, and softplus nonlinearities after each layer (except the last). For tensor-valued sources, we use standard source-specific transforms (e.g., CNNs for images). Unless mentioned otherwise, we use the RealNVP (Dinh et al., 2017) normalizing flow (NF) for the density model $p_{\boldsymbol{y}}$ (used for variable-rate and fixed-rate LTC), with 5 coupling layers and isotropic Gaussian prior. The rates plotted (unless mentioned otherwise) are given by the cross-entropy of the density model. We use 4096 samples for the Monte-Carlo estimate of the PMF in (5) and (8) for all lattices. This was sufficient for most sources and lattices; see Appendix. A.2. In most experiments, we compare with ECVQ as well as the rate-distortion function $R(D)$ of the source. When not known in closed-form, $R(D)$ is estimated using Blahut-Arimoto (BA) (Blahut, 1972; Arimoto, 1972) for 1-D and 2-D sources, and NERD (Lei et al., 2022) for higher-dimensional sources.

### 5.1 LTC AS A ONE-SHOT CODE

We first consider 2-D isotropic Gaussians and the 2-D Banana source, where we can easily visualize the quantization regions. Then, we consider $n$-length i.i.d. sequences of several sources, including Gaussian with $L_2$ distortion, Laplacian with $L_1$ distortion, and a 1-dimensional marginal of the Banana source (Ballé et al., 2021) with $L_2$ distortion. The 1-d Banana marginal is used to test a skewed distribution, since both Gaussian and Laplacian sources are symmetric. In this setting, $n$ is the dimension of both $\boldsymbol{x}$ and the latent space $\boldsymbol{y}$. We found that using STE for the rate term worked better for the i.i.d. sources, whereas dithering was more stable for the Banana source.

**2-D Gaussian and Banana.** In 2-dimensions, we choose $Q_\Lambda$ to be the hexagonal lattice, which is known to be optimal (Conway & Sloane, 1999), and use the NF density model. As shown in Figs. 5, LTC produces quantization regions in the original source space that closely match that of optimal ECVQ in Fig. 3. The latent space (Fig. 17b) effectively uses the hexagonal lattice, compared to NTC's use of the integer lattice (Fig. 17a). The analysis transform $g_a$ determines which source realizations $\boldsymbol{x}$ get quantized to a specific lattice point in the latent space (colored regions); the synthesis transform $g_s$ maps the lattice points to their corresponding reconstructions in the original space (black dots). Shown in Fig. 6, the LTC rate-distortion performance matches that of ECVQ for the Gaussian. On the Banana, NTC and LTC achieve ECVQ at low rates; at larger rates, LTC improves on NTC but does not fully match ECVQ; see Fig. 6. Inspecting the quantization regions (Fig. 19, 20, 21) reveals why. At low rates, all methods quantize along a 1-D nonlinear component of the source, resulting in the same quantization regions no matter the tessellating structure. At higher rates, 2 dimensions are used for quantization, where the integer lattice of NTC becomes sub-optimal compared to the hexagonal regions of ECVQ/LTC. We see that the quantizer used solely determines LTC's quantization regions; whereas the transforms' purpose is to warp the latent quantizer onto the source statistics.

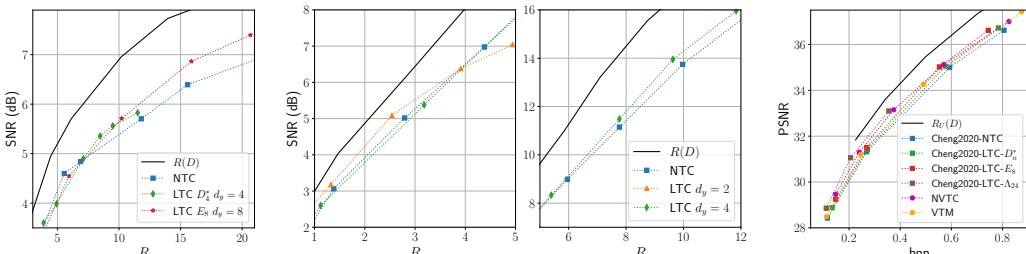

Figure 7: LTC on real-world sources. *Left:* Speech. *Middle two*: Physics. *Right*: Kodak.

**$n$-length i.i.d. Sequences.** Here, we choose $Q_\Lambda$ to be the best-known lattices in $n$-dimensions, including $A_2$ (hexagonal), $D_4^*$, $E_8$, and $\Lambda_{24}$. As $n$ grows, ECVQ performance will approach the rate-distortion function of the source, which for the Gaussian (squared error distortion) is given by $R(D) = \frac{1}{2}\log\left(\frac{\sigma^2}{D}\right), 0 \le D \le \sigma^2$ and the Laplacian (absolute error distortion) is given by $R(D) = -\log(\lambda D), 0 \le D \le 1/\lambda$ (Cover & Thomas, 2006; Equitz & Cover, 1991). On Gaussians, we see that in Fig. 6, LTC performs very close to optimal ECVQ in dimensions $n = 2, 4, 8$ using the hexagonal, $D_n^*$, and $E_8$ lattice respectively. For larger $n$, ECVQ is too costly to compute, but we see that LTC with the Leech lattice for $n = 24$ continues the trend, further improving the rate-distortion performance and approaching $R(D)$. We observe similar performance gains for the Laplacian and Banana-1D in Fig. 15. Since each of the lattice quantizers used have efficient CVP algorithms, LTC succeeds in larger dimensions and rates, whereas ECVQ could not even be run due to its exponential complexity. This result provides evidence that companding (in the form of LTC) is near-optimal in terms of VQ performance, even if it may not exactly implement VQ (Gersho, 1979).

**Real-World Sources.** We now consider sources $\boldsymbol{x}$ drawn from real-world data. We consider the Physics and Speech datasets (Yang & Mandt, 2022), of dimension $d = 16$ and $33$ respectively, and the Kodak image dataset (Franzen), which has variable dimension depending on image size.

For Physics and Speech, we map $\boldsymbol{x}$ to a latent space with dimension $d_y$ using MLP-based transforms. For these sources, we found training instabilities with STE, so dithering is used instead. At higher rates, we also observed stabler training with the factorized density model compared to the normalizing flow. For Speech, we observe coding gain for rates above $\approx 7$ bits with the $D_4^*$ and $E_8$ lattices, shown in Fig. 7. Coding gain is observed on Physics for LTC in different rate regimes. LTC with the hexagonal lattice performs best at lower rates, but the performance drops off as rate increases. At higher rates, LTC with 4 latent dimensions performs the best with $D_4^*$ lattice. These rates are measured in bits-per-sample, and not normalized by dimension. Generally speaking, different rate regimes may require a different number of latent dimensions to be used for NTC and LTC. If the number of latent dimensions used by the transforms does not match the lattice quantizer dimension, this can result in suboptimal performance as the tessellation in source space may not be optimal. This is likely why LTC performance "peaks" for different latent dimensions.

For large-scale images, we focus on the NTC architecture in Cheng et al. (2020), and use product $E_8$ and Leech $\Lambda_{24}$ lattices along the channel dimension of the latent. We use dither during training and STE at test time, and train on the Vimeo-90k (Xue et al., 2019) dataset. We use the identical hyperparameter settings for transforms and entropy models, corresponding to the first 4 quality levels for the "cheng2020-attn" model in CompressAI (Bégaint et al., 2020). Shown in Fig. 7, LTC with product $E_8$ and product Leech achieves a -5.274% and -16.708% BD-rate gain respectively over NTC when evaluated on Kodak. In Sec. B, we compare all methods with Cheng2020-NTC and VTM as anchor in the BD-rate table (Tab. 1). While NVTC achieves a -2.919 BD-rate gain over VTM, Cheng2020 with Leech is significantly better, with a -11.895% BD-rate gain. Note that NVTC and Cheng2020 models differ significantly in terms of the transforms and entropy models, so multiple factors (e.g., transforms and quantization) may contribute to the achieved performance. Additionally, NVTC applies vector quantization to vectors of size 16, meaning NVTC is able to leverage the packing efficiency of 16-dimensional space compared to the $E_8$ and Leech with 8 and 24 respectively. To compare with LVQAC, we additionally plot the performance using the $D_n^*$ lattice, which achieves a -2.337% BD-rate gain over Cheng2020-NTC. This compares *solely* the effect of quantization scheme, and not other components of LVQAC (Zhang & Wu, 2023). This result, along with how the $E_8$ and $\Lambda_{24}$ perform relative to each other, further supports evidence that the rate-distortion performance is

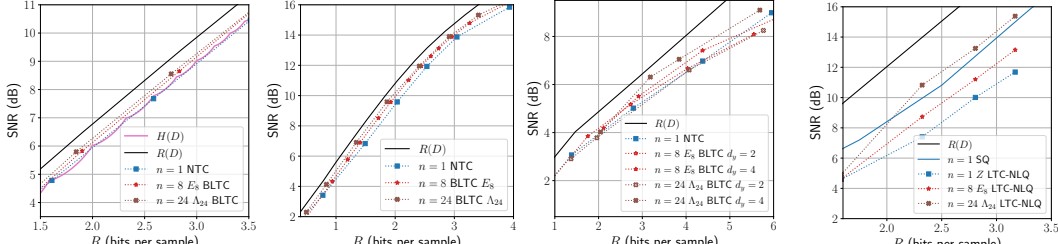

Figure 8: Block coding. *Left*: Sawbridge. *Center*: Banana. *Right*: Physics. Figure 9: LTC-NLQ.

significantly determined by the quantization choice in the latent space, and that the NTC model is able to leverage the enhanced packing efficiency of higher-dimensional lattices. We see that simply adding lattice quantization to the latent space of Cheng2020 has brought its performance to approach the ResNet-VAE rate-distortion upper bound $R_U(D)$ from Yang & Mandt (2022) on Kodak. To compare just the speed of SQ vs LQ, we run a forward pass through the transforms, quantization, and entropy model. When run on single-core CPU, Cheng2020-NTC takes 3.241 seconds per image averaged over Kodak, whereas Cheng2020-LTC-E8 takes 4.002 seconds per image. Additional comments on complexity can be found in Sec. A, B, with full runtime comparison in Tab 2.

As NTC and LTC operate here as one-shot compressors of $x$, these results demonstrate that the gap to $R(D)$ is partially due to NTC being sub-optimal as a one-shot code. Fully approaching $R(D)$, however, requires encoding across multiple source realizations, which we discuss in Sec. 5.2.

**Nested Lattice LTC.** Here, we demonstrate the ability of fixed-rate LTC with nested lattices to provide coding gain with lattice dimension, with a much lower complexity when coding to bits. On i.i.d. Gaussians (Fig. 9), we demonstrate LTC with NLQ in the latent space. We use self-similar nested lattices for the integer, $E_8$ and $\Lambda_{24}$ lattices, and set the nesting ratio $\Gamma = 5, 7, 9$ to sweep the rate. The performance improves as the lattice dimension grows, and approaches $R(D)$ as before, but does not perform as well as variable-rate LTC for a particular dimension, as expected. We note that LTC with NLQ has encoding complexity linear in $n$ and is able to outperform NTC.

### 5.2 Block Coding

We evaluate BLTC in the block coding setup (i.e., compressing multiple samples from dataset simultaneously) on the Sawbridge, Banana, and Physics sources. For the Sawbridge, the optimal VQ (i.e., one-shot coding) performance is known in closed form as $H(D)$ (Wagner & Ballé, 2021, Thm. 2). In this case, BLTC uses a latent dimension $d_y = 1$, and the nominal dimension is $d = 1024$. In Fig. 8, we verify that BLTC bridges the gap between $H(D)$ and $R(D)$. Note that going beyond $H(D)$ is impossible without compressing multiple realizations at time; this verifies the ability of BLTC to leverage space-packing of the Sawbridge's latent source. For the Banana and Physics sources, the optimal one-shot performance is unknown, but we see that BLTC improves upon NTC and LTC, which are both one-shot codes. Thus, we show BLTC's ability to approach $R(D)$.

### 5.3 Ablation Study

We first show how it is necessary for a good lattice to be used to achieve optimal performance. Only using the best-known lattices in each dimension yielded near-ECVQ performance. If sub-optimal lattices are used, this was not the case. Fig. 16a shows the $A_n$ and $D_n^*$ lattices used for $n = 8$; they perform about the same as ECVQ for $n = 4$, but still yield coding gain over NTC. This implies that LVQAC, which uses $D_n^*$ is sub-optimal compared to other lattices. In the appendix (Sec. B.1), we provide several additional ablations further revealing the interplay of LTC components.

## 6 Discussion and Limitations

In this work, we use the best known lattices up to dimension 24. Going to even larger dimensions will improve performance further; designing efficient high dimensional lattices with low complexity CVP solvers is an active area of research building on advances in modern coding theory (e.g. low density parity codes and polar codes).

## ACKNOWLEDGEMENTS

This work was supported by The Institute for Learning-enabled Optimization at Scale (TILOS), under award number NSF-CCF-2112665. The work of Hamed Hassani was further supported by NSF CAREER award CIF-1943064, and the work of Eric Lei was further supported by a NSF Graduate Research Fellowship.

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

# A EXPERIMENTAL DETAILS

The code for these experiments will be released upon publication. In all experiments, we use a batch size of 64 and train until convergence using Adam. For the i.i.d. scalar sequences, we found that using CELU nonlinearities with no biases in the transforms sometimes helped improve training stability. For synthetic sources, Speech and Physics, the rate-distortion Lagrange multiplier $\lambda$ is swept over 0.5, 1, 1.5, 2, 4, 8. For images, we use the default $\lambda$ values in Bégaint et al. (2020). In addition, for images, a product lattice is applied along the channel dimension of the latent tensor, which has shape $C \times H \times W$. There are $C/n$ lattices applied $H \cdot W$ times, where $n$ is the lattice dimension.

## A.1 CVP ALGORITHMS

In this paper, we use the CVP algorithms outlined in Conway & Sloane (1999, Ch. 20). For the Barnes-Wall lattice $\Lambda_{16}$ in dimension 16, we use the algorithm outlined in Conway & Sloane (1984). Finally, the Leech lattice $\Lambda_{24}$ in dimension 24 has the slower method described in Conway & Sloane (1984) and the improved version in Conway & Sloane (1986). In many cases, the lattice $\Lambda = \bigcup_{i=1}^{T}(r_i + \Lambda_0)$ is a union of $T$ cosets of a sublattice $\Lambda_0$, and one can solve CVP for $\Lambda$ by computing $Q_\Lambda(x) = \arg\min_{1 \leq i \leq T} Q_{\Lambda_0}(x - r_i) + r_i$. For example, the $E_8$ lattice is equivalent to two cosets of $D_8$.

## A.2 MONTE-CARLO SAMPLES

Regarding variable-rate LTC, for lattices of lower dimensions, we found that setting the number of Monte-Carlo samples $N_{\text{int}}$ for approximation of (5) to be 4096 was sufficient to recover near-ECVQ performance. For the Leech lattice of dimension 24, however, better performance could be recovered by increasing $N_{\text{int}}$ at the cost of increased GPU memory requirements; see Fig. 10.

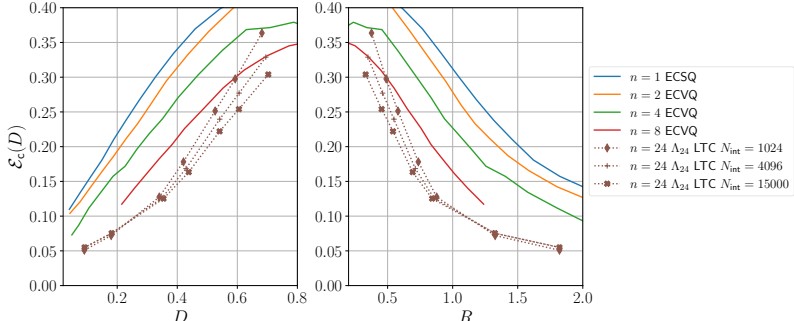

Figure 10: Effect of $N_{\text{int}}$ on the Leech lattice, for i.i.d. Gaussian sequences.

We see that setting $N_{\text{int}} = 1024$ results in worse performance than ECVQ for $n = 8$ at low rates. Setting $N_{\text{int}} = 4096$ improves the performance, and nearly approaches that of $N_{\text{int}} = 15000$.

## A.3 SOURCE GENERATION

The Gaussian and Laplacian sources are straightforward to simulate. The Banana source is taken from the Tensorflow Compression library (Ballé et al., 2024). To generate the Banana-1d marginal, we simply take the first dimension of the 2-d Banana realizations. The Sawbridge source (Wagner & Ballé, 2021) is a continuous random process

$$X(t) = t - \mathbb{1}\{t \geq U\}, \quad 0 \leq t \leq 1, \tag{10}$$

where $U \sim \text{Unif}([0, 1])$. Following Wagner & Ballé (2021), we uniformly sample 1024 time steps between 0 and 1, which creates 1024-dimensional vectors. For all synthetic sources, we sample $10^7$ samples as our training dataset. Finally, the Physics and Speech sources are taken from https://github.com/mandt-lab/RD-sandwich.

## A.4 EFFICIENCY

Compared to NTC, LTC requires more memory/training time, roughly 50% more for the low/moderate dimension sources and 10% more for images. The main computational bottleneck that LTC has is the use of Monte-Carlo integration for (5). Sampling uniformly random vectors from the Voronoi region of a lattice requires a quantization step, and hence solving the CVP. Since we use 4096 sampled vectors per batch, this means CVP needs to be solved on all 4096 vectors. The lattices chosen above all have GPU-friendly implementations that can parallelize across the input vectors. On NVIDIA RTX 5000 GPUs with 16 GB memory, LTC training until convergence took at most a few hours for the Speech and Physics sources, and minutes for the i.i.d. scalar sequences. This was roughly twice the amount of time to train NTC. However, on large-scale images, the training speed is dominated by the backwards pass through the convolutional neural networks, and so LTC takes around the same time as NTC (around 10 days per rate point).

## B ADDITIONAL RESULTS

**BD-rate and runtime tables for Kodak.** Here, we show the BD-rate tables (Tab. 1) and full runtime comparison (Tab. 2) on the Kodak benchmark. For the runtime comparison, it is time taken per-image on a single-core CPU, averaged over the rate-distortion tradeoff. We see that Cheng2020-LTC-$\Lambda_{24}$ outperforms all other methods in rate-distortion performance, including ELIC (He et al., 2022). We note that the ELIC model is NTC-based, and thus even further gains can be expected by integrating LTC with ELIC transforms and entropy model. This would also inherit the efficient nature of the ELIC layers, as the quantization and entropy modelling are independent components of the neural compressor. We leave this for future work as it is out of the scope of the current study. Regarding runtime on the Cheng2020 architecture, using the $E_8$ lattice incurs a 1.23x increase in runtime over NTC, whereas the Leech lattice incurs a 2.68x increase. We note that the Leech lattice CVP solver we used (Conway & Sloane, 1986) is not the fastest; one can expect faster runtimes with the Leech decoder of Vardy & Be'ery (1993).

Table 1: BD-rate comparison on Kodak.

| Model | BD-rate (NTC anchor) | BD-rate (VTM anchor) |
|---|---|---|
| Cheng2020-NTC | 0.000 | 5.210 |
| Cheng2020-LTC-$D_n^*$ | -2.337 | 2.753 |
| Cheng2020-LTC-$E_8$ | -5.274 | -0.335 |
| Cheng2020-LTC-$\Lambda_{24}$ | -16.708 | -11.895 |
| NVTC (Feng et al., 2023) | -4.953 | -2.919 |
| VTM (Bross et al., 2021) | -7.879 | 0.000 |
| ELIC (He et al., 2022) | -10.607 | -5.656 |

Table 2: Inference encoding runtime on Kodak, single CPU. LTC models use 1024 samples for Monte-Carlo estimation.

| Model | Runtime (sec.) |
|---|---|
| Cheng2020-NTC | 3.241 |
| Cheng2020-LTC-$D_n^*$ | 4.016 |
| Cheng2020-LTC-$E_8$ | 4.002 |
| Cheng2020-LTC-$\Lambda_{24}$ | 8.634 |
| ELIC (He et al., 2022) | 3.132 |

**NTC with lattice quantization during inference.** Here, we show the performance of LTC when the transforms and entropy model are loaded from a pre-trained NTC model. In other words, a trained NTC model's scalar quantizer is replaced by a lattice quantizer at inference time. The performance of this scheme is largely dependent on the source at hand. Fig. 11 shows that on the Gaussian source, this scheme performs the same as training LTC from scratch; it may shift the point on the R-D tradeoff but overall lies on the same R-D curve. This is perhaps not unexpected, as the transforms learned by NTC and LTC for the i.i.d. Gaussian source effectively perform a scaling of the source, and so interchanging them has no effect. We use the same Flow entropy model for fair comparison. In general, however, this is not expected to be the case, as the analysis transform determines which

region of the lattice gets used; for more complex sources, there may be more sophisticated ways the transform determines how the lattice is used. For example, in Fig. 12, we demonstrate the same scheme on Kodak images with the Cheng2020-LTC-$\Lambda_{24}$ model. Here, we see that while using a Leech lattice quantizer with the trained NTC model improves the performance over NTC by a BD-rate of -12.984%, training the model with the lattice quantizer helps boost the performance further to a BD-rate of -16.708%. This result implies that one can avoid training from scratch when a good set of transforms and entropy model are already trained from a NTC architecture, and instead potentially fine-tune the NTC transforms and entropy model with the chosen lattice for best performance. In the future, when better transforms and entropy models are developed, one can also train with the lattice from scratch to get the best performance, as the training times between NTC and LTC (from scratch) are on the same scale (weeks).

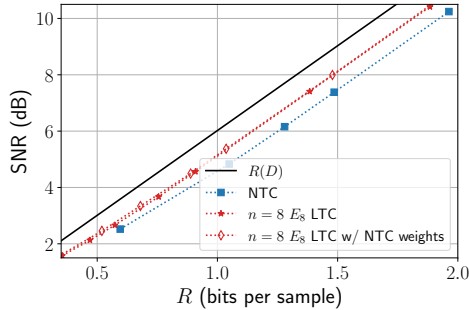 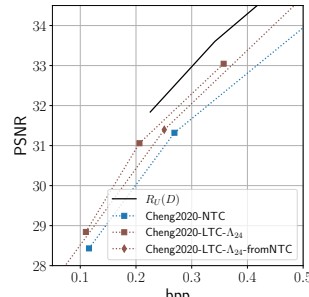

Figure 11: Gaussian source, comparing LTC (trained) versus LTC loaded with pretrained NTC weights.

Figure 12: Kodak images, comparing LTC (trained) versus LTC loaded with pretrained NTC weights.

**Learned quantizer cells in regions of low probability.** In Figs. 13, 14, we visualize the quantizer cells of LTC on regions of low probability density, for the 2D Gaussian source and Banana source, respectively. That is, we train LTC on a source $p_{\boldsymbol{x}}$, and evaluate on the support of $p_{\boldsymbol{x}}$ as well as regions outside the support of $p_{\boldsymbol{x}}$. Overall, the quantization cells successfully generalize outside the support of $p_{\boldsymbol{x}}$, meaning that if a sample $\boldsymbol{x}$ to be compressed is perturbed slightly outside the support of $p_{\boldsymbol{x}}$, its reconstruction will still be close to $\boldsymbol{x}$.

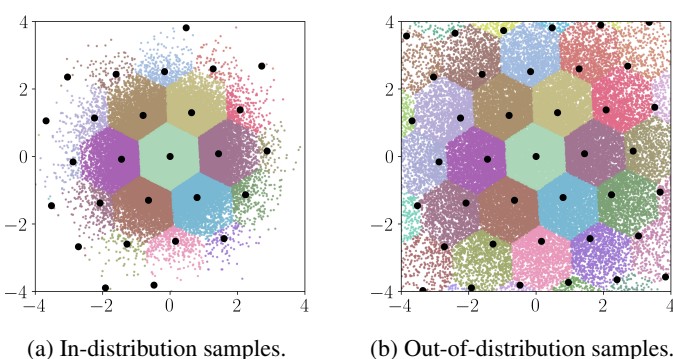

(a) In-distribution samples.  (b) Out-of-distribution samples.

Figure 13: Quantizer cells of LTC trained on 2-d i.i.d. Gaussians.

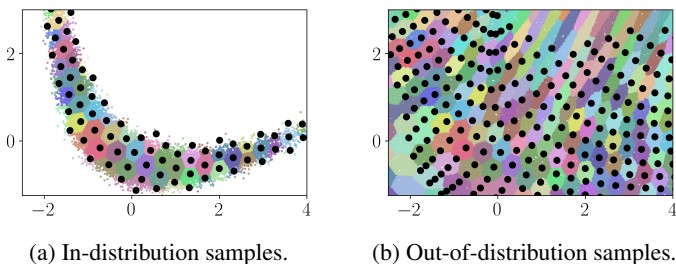

(a) In-distribution samples.  (b) Out-of-distribution samples.

Figure 14: Quantizer cells of LTC trained on Banana source.

**Rate-redundancy.** For certain figures in the ablation study, we define the *normalized rate-redundancy* of a compressor c,

$$\mathcal{E}_{\mathsf{c}}(D) := (R_{\mathsf{c}}(D) - R(D))/R(D), \tag{11}$$

where $R_{\mathsf{c}}(D)$ is the rate achieved by c at distortion $D$, and $R(D)$ is the rate-distortion function of $S$. Intuitively, $\mathcal{E}_{\mathsf{c}}(D)$ measures the fraction of additional bits per sample required to code a sequence relative to $R(D)$, and makes some of the performance gains easier to see in the figures.

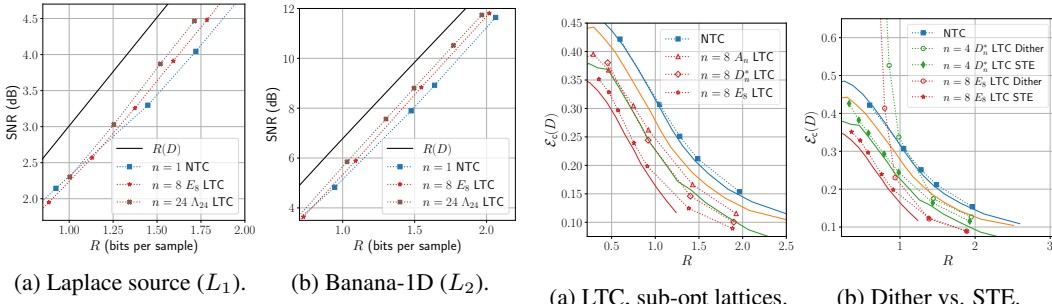

(a) Laplace source ($L_1$).  (b) Banana-1D ($L_2$).

Figure 15: Rate-distortion performance of scalar i.i.d. sequences.

(a) LTC, sub-opt lattices.  (b) Dither vs. STE.

Figure 16: Ablation study, i.i.d. Gaussians.

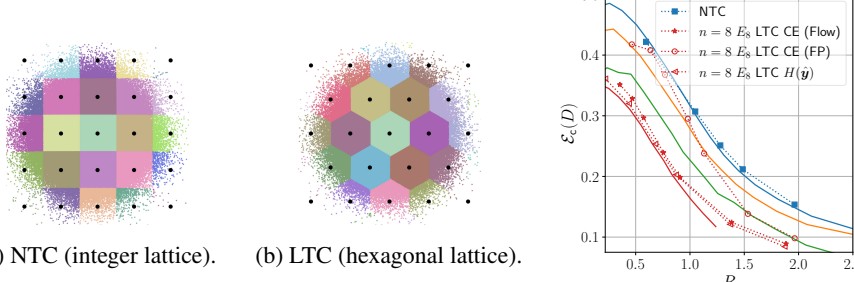

(a) NTC (integer lattice).  (b) LTC (hexagonal lattice).

Figure 17: Latent space quantization regions.

Figure 18: Flow provides tighter entropy estimate.

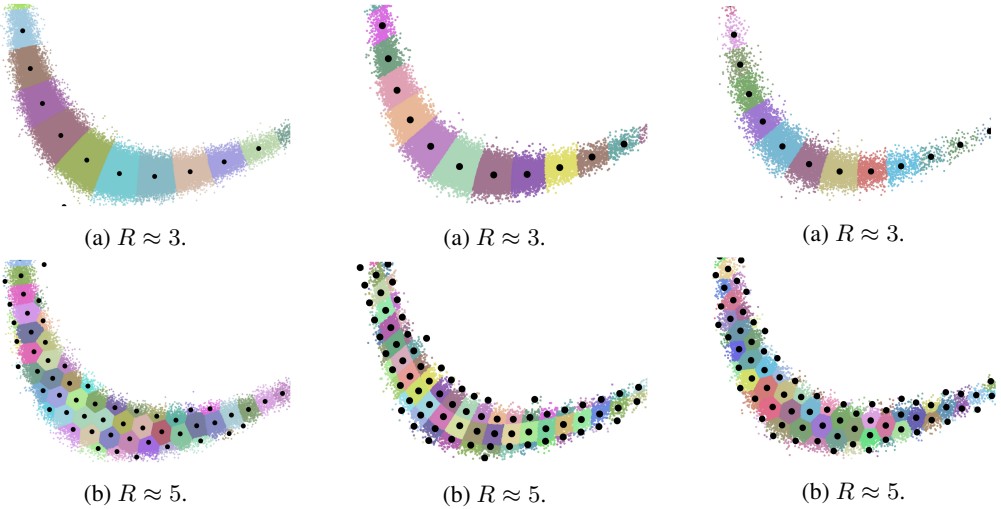

(a) $R \approx 3$.  (a) $R \approx 3$.  (a) $R \approx 3$.

(b) $R \approx 5$.  (b) $R \approx 5$.  (b) $R \approx 5$.

Figure 19: ECVQ quantization regions, banana source.

Figure 20: NTC quantization regions, Banana source.

Figure 21: LTC quantization regions, Banana source.

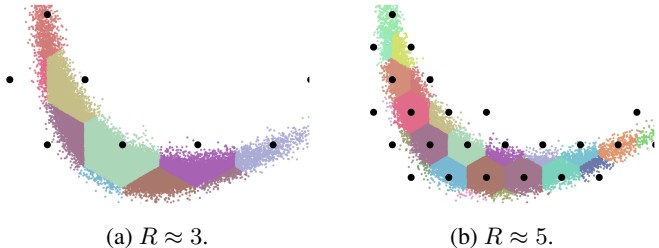

(a) $R \approx 3$.                    (b) $R \approx 5$.

Figure 22: Quantization regions of Banana source using ECLQ. Without the transforms, quantization regions of ECLQ fail to conform to the source manifold.

## B.1    ABLATION STUDY

We first provide a comparison with entropy-constrained lattice quantization (ECLQ), which applies lattice quantization directly to the source without transforms. While ECLQ approaches VQ performance in the large rate regime, it is highly sub-optimal in the low/moderate rate regime. LTC, on the other hand, performs much better at all rates. This is shown in Fig. 23 for Gaussians and Banana. While the quantization regions look similar, the primary benefit of LTC is that the $g_s$ transform moves the reconstructions $\hat{x}$ according to the source statistics. Standard ECLQ keeps the reconstructions at the center of the lattice cells, which is sub-optimal for non-uniform sources. Moreover, on the Banana source, without the transforms, ECLQ fails to match the quantizer cells to the source statistics (Fig. 22), unlike LTC (Fig. 21). We further analyze (i) the training objective used, and (ii) the effect of density models on the entropy estimate of $\hat{y}$. For the training objective used, we compare using dithered quantization with STE for the rate term (i.e., using likelihoods from (6) and (5) respectively). Shown in Fig. 16b, for low rates, the reported cross-entropies from dithering are much larger than that of STE, but converge to the STE performance as rate increases. One possible reason is that the convolved density in (6) over-smooths the true density of $y$ in the low-rate regime, leading to sub-optimality.

For the entropy models, we compare the factorized density with the normalizing flow density in Fig. 18. Both models use STE during training. Interestingly, when computing the true entropy of the quantized latent $H(\hat{y})$, both models yielded near-optimal performance (curve labeled $H(\hat{y})$). However, the cross-entropy upper bound of the factorized density was not as tight as that of the normalizing flow. The normalizing flow has a near-zero gap between $H(\hat{y})$ and the cross entropy.

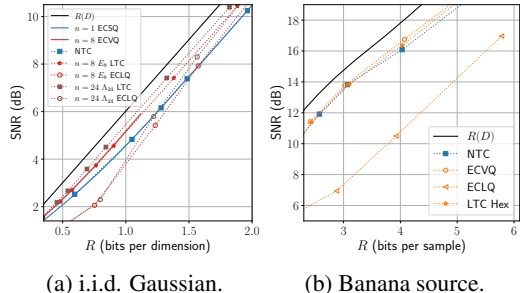

(a) i.i.d. Gaussian.                    (b) Banana source.

Figure 23: LTC vs. ECLQ.

