# OpenReview forum: "Approaching Rate-Distortion Limits in Neural Compression with Lattice Transform Coding"
_ICLR.cc/2025/Conference — ICLR 2025 Spotlight_

### Official Review · Reviewer_bzaC · 2024-10-27

**Soundness:** 3
**Presentation:** 2
**Contribution:** 2
**Rating:** 6
**Confidence:** 4

**Summary:**

This paper highlights the limitations of scalar quantization in neural compression and introduces lattice transform coding as a solution. Given the challenges in rate estimation within this approach, the authors propose a Monte Carlo-based method to address it. To manage complexity, they further introduce a nested lattice structure. Experimental results are presented using both synthetic datasets and real-world datasets, including speech and the Kodak image dataset.

**Strengths:**

The observation that neural transforms alone cannot overcome scalar quantization limitations is a valuable insight.
The use of dithered noise and Monte Carlo estimation for rate estimation is also noteworthy.

**Weaknesses:**

- The presentation could be improved for clarity. All figures are too small, and Section 3 (on the suboptimality of scalar quantization) would benefit from additional detail beyond the currently limited Figure 2.
- Additionally, the distinction in block LTC needs clarification.
- Lattice coding encoding complexity remains an issue (except for $n=24$). More specifics on how this scheme was applied to the Kodak image dataset would be helpful.

**Questions:**

See weaknesses.

---

> ### Author Response · Authors · 2024-11-21
>
> Thank you for taking the time to read the paper and provide valuable feedback! Please find our responses to your comments in the following point-by-point response, which refers to the revised manuscript.
>
> **Response to weaknesses**:
>
> **W1**. We have improved the clarity of some of the writing, such as more detailed comparison of the methods for image compression, clarifying some of the motivation for nested LQ and block LQ, the role of lattice quantizer dimension, and hyperparameters such as $\lambda$, as suggested by other reviewers as well. Regarding Sec. 3, we have added additional details at the end of the section to reflect the larger gap between NTC and optimal performance as the dimension grows, to further support the suboptimality of scalar quantization.
>
> **W2**. Thanks for your suggestion, and we have clarified the block LTC section (4.3) accordingly. To clarify, neural compression has traditionally been designed for images, which naturally results in a one-shot compressor design (compresses one source vector at a time). In information theory, it is known that compressing multiple i.i.d. realizations simultaneously achieves optimality and outperforms one-shot compressor designs. This can be useful in practice for certain settings such as dataset compression, e.g. when one wishes to compress a dataset of scientific measurements or audio samples, where each sample is independent, like the Physics and Speech datasets. In the neural compression literature, no such block coding scheme currently exists. Our proposed block LTC method provides a low-complexity framework to realize block coding, by building on LTC to apply LQ across iid latent vectors. To put things in perspective, the LTC framework recovering VQ and achieving optimality for vector source in a one-shot sense is the major contribution of our work, and block LTC is an extension of our LTC framework, showing that one can approach rate-distortion limits in block coding.
>
> **W3**. Regarding complexity, the total complexity for a neural compressor is determined by the transform, likelihood computation, quantization of latents, and entropy coding. On images, we use the same transforms and autoregressive entropy model as the Cheng2020 model, and it is well-known that the complexity of neural compressors is dominated by the autoregressive entropy model [1] and not quantization. For quantization, the complexity of solving CVP is not an issue. For each lattice, CVP can be computed at any rate in a fixed number of operations, and is significantly faster than the codebook search in VQ, which is exponential in the rate. We do not evaluate entropy coding in this paper as it is not the focus of our paper, and we leave this to future work; as mentioned in Sec. 5, the rates reported are the cross-entropies of the learned entropy model. However, we do not expect this to increase complexity significantly for the lattices used in this work. For example, for the $E_8$ lattice used for image compression (Fig. 7), the effective codebook size will be around $2^(8*\text{bpp})$, which is a codebook size commonly seen in reduced-complexity VQ methods such as NVTC [2], and there are also 8x less symbols to code compared to scalar quantization. Additionally, Sec. 4.2 discusses the use of nested LQ, which provides a low-complexity solution to encoding with lattices. Regarding how LTC is applied for image compression, we have added details in the manuscript. To summarize, we apply lattices across the channel dimension of the latent tensor of the Cheng2020 model, which is usually either 128 or 192, depending on rate. For the $D_n^*$ lattice, it is defined for arbitrary dimension, so we can directly apply it to the channel dimension. For the E8 and Leech lattices, they are only defined for 8 and 24 dimensions respectively, so we use product lattices. For the E8 lattice, this results in 16 or 24 E8 lattices, and for the Leech lattice, this results in 5 or 8 Leech lattices (for the former, we use a channel dimension of 120 instead of 128). We have updated the experimental details in the revised draft to incorporate these details.
>
> References:
>
> [1] Y. Yang, S. Mandt, and L. Theis. An introduction to neural data compression. arXiv preprint arXiv:2202.06533, 2022.
>
> [2] Runsen Feng, Zongyu Guo, Weiping Li, and Zhibo Chen. Nvtc: Nonlinear vector transform coding.
> In Proceedings of the IEEE/CVF Conference on Computer Vision and Pattern Recognition, pp.
> 6101–6110, 2023.

---

> > ### Comment · Reviewer_bzaC · 2024-11-26
> >
> > My concerns have been adequately addressed in the authors’ responses and reflected in the revised manuscript. The additional clarity provided on the suboptimality of scalar quantization, the explanation of block LTC, and the improved experimental details have enhanced the soundness and presentation of the paper. As a result, I have raised my original score to reflect these improvements.

---

> > > ### Author Response · Authors · 2024-11-26
> > >
> > > Thank you! We appreciate your engagement during the discussion period and again for the constructive feedback that has improved the paper.

---

### Official Review · Reviewer_WNyg · 2024-11-03

**Soundness:** 4
**Presentation:** 4
**Contribution:** 3
**Rating:** 8
**Confidence:** 4

**Summary:**

The authors show the suboptimality of existing neural compression models for simple multidimensional sources, such as 2-D Gaussians. They point out that the scalar quantization used in current neural compression models can lead to the suboptimality. They then propose a solution, *i.e.*, lattice transform coding, with lattice quantization in latent space. Extensive experiments verify the superiority of the proposed lattice transform coding.

**Strengths:**

1. The idea of using lattice quantization in neural compression has potential whose benefits cannot be achieved by improving transforms and the entropy model.
2. The theoretical analysis and experimental evaluation presented in this paper are convincing. The authors have carried out extensive experiments on both synthetic sources and on higher dimensional real sources.
3. The paper is well written and easy to follow.

**Weaknesses:**

1. The real-world application of the proposed Lattice Transform Coding (LTC) may be limited by the computational complexity of training and inference.

   - *On training.* The authors show that training an LTC model per rate point for image compression takes about 10 days. By comparison, training of nonlinear transform coding (NTC) models can be reduced by fine-tuning the lower rate model from a pre-trained higher rate model. I'm not sure if such tricks can also speed up the training process of LTC.
   - *On inference.* The authors provide a comparison of the NTC method (Cheng2020-NTC) with the proposed LTC method (Cheng2020-LTC-E8), which shows that the computational complexity of LTC is comparable to NTC. However, Cheng2020-NTC is a nonlinear transform coding method using spatial auto-regression in entropy modeling, which is notoriously time consuming. The method is further refined in subsequent work, such as wavefront coding, checkerboard context, and channel-wise auto-regression. Notably, the proposed lattice transform coding can be even more computationally expensive than spatial auto-regression. This can be a fatal flaw in applications.

2. Typos:

   - Line 470. The citation of Vimeo-90k should be realized by a `\citep` command instead of `\citet`.
   - Line 472. The citation of CompressAI is not included in the reference.

**Questions:**

1. Is it possible to further reduce the computational complexity of LTC models for training and inference? Authors can try the following things to reduce concern.
   - Provide more detailed computational complexity analysis, comparing LTC to state-of-the-art NTC methods beyond just Cheng2020-NTC.
   - Discuss potential approaches to reduce training and inference time for LTC, such as whether fine-tuning techniques used for NTC could be adapted.
   - Comment on how the complexity of LTC scales with different lattice dimensions and source dimensionality.
   - Address whether there are any trade-offs between computational complexity and rate-distortion performance for LTC.
2. Figure 5 does not show the learned lattice of LTC in the regions of lower probability density. How does LTC perform in these regions? Please provide visualizations or quantitative analysis of LTC's performance in lower probability density regions, and discuss any potential implications for overall compression performance.
3. LTC provides a construction on the quantization that is quite different from previous work. Then, how do the nonlinear transforms affect the rate-distortion performance of the LTC models? Please provide an ablation study or analysis that isolates the effects of the nonlinear transforms on LTC's rate-distortion performance.

---

> ### Author Response · Authors · 2024-11-21
>
> Thank you for taking the time to read the paper and provide valuable feedback! Please find our responses to your comments in the following point-by-point response, which refers to the revised manuscript.
>
> **Response to weaknesses**:
>
> **W1**. Regarding training, fine-tuning a pre-trained model (such as a lower rate one) can indeed be done for LTC. Architecturally, the parameters that are learned are the same (i.e., transforms and entropy model); the only difference is the choice of quantization, which is not learned. On inference: we acknowledge the reviewer’s comments that spatial auto-regression is time consuming. However, the use of lattice quantization is independent of the choice of entropy model. The Cheng2020 architecture is not our contribution; we merely use it to compare how scalar quantization and lattice quantization perform for that particular architecture. The LTC framework can indeed be used with other NTC architectures, such as those with more refined entropy models like those you mentioned. As mentioned in Sec. 5.1, and the new runtime tables in Appendix B (Table 2), we compare just the effect of lattice quantization for runtime efficiency during inference on Cheng2020, and show that using the $E_8$ lattice (with dimension 8) incurs a 1.23x increase in runtime over NTC, whereas the Leech lattice (with dimension 24) incurs a 2.68x increase. Thus, the use of lattice quantization is not significantly more computationally expensive than using scalar quantization with the spatial autoregression.
>
> **W2**. Thanks for catching the typos and missed reference. We have updated the paper accordingly.

---

> > ### Author Response · Authors · 2024-11-21
> >
> > **Response to questions**:
> >
> > **Q1**: Regarding complexity:
> >
> > - As mentioned in the response to weakness 1, we have added a runtime table in Table 2 in Appendix B, where we compare just the effect of lattice quantization for runtime efficiency during inference on Cheng2020, and show that using the $E_8$ lattice incurs a 1.23x increase in runtime over NTC, whereas the Leech lattice incurs a 2.68x increase. In addition we show the runtime for ELIC, which has a more advanced entropy model than Cheng2020, and is slightly faster than Cheng2020. Comparing with the BD-rate table in Table 1, we see that Cheng2020-LTC with Leech lattice has a higher inference complexity (2.76x over ELIC), but yields a -11.895\% BD-rate improvement over VTM compared to ELIC’s -5.656\% BD-rate improvement (which is the second best). This is despite ELIC having more advanced transforms and entropy models; note that Cheng2020-NTC has slightly slower runtime to ELIC but significantly worse BD-rate (5.210\% worse than VTM). This result highlights the importance of quantization choice, in that it can boost performance independently of architecture design. Thus, it appears that combining ELIC with LTC would combine the best of both worlds and yield even better performance-complexity tradeoffs; we leave this for future work.
> >
> > - Regarding fine-tuning approaches to reduce complexity such as those used in NTC, these can be similarly applied to LTC. As mentioned in response to weakness 1, the lattice quantizer has no learned parameters, so any learned module of NTC can be similarly trained or fine-tuned for LTC. In addition, in Appendix B, we have added Fig. 12, demonstrating that by taking a trained Cheng2020-NTC model, and applying the Leech lattice, we already get a performance gain over Cheng2020-NTC. While using a Leech lattice quantizer with the trained NTC model improves the performance over NTC by a BD-rate of -12.984\%, training the model with the lattice quantizer helps boost the performance further to a BD-rate of -16.708\%. This suggests that we can likely fine-tune a LTC model initialized with the transforms and entropy model from a pretrained NTC model to achieve good performance with less training.
> >
> > - For a single lattice, the runtime to lattice-quantize is a fixed number of operations that the CVP solver requires, and is independent of the rate or input to the CVP solver. Since it is a fixed number of operations, it is significantly less than the operations required for a VQ codebook search, which grows exponentially in the dimension and rate. The complexity of a forward-pass through LTC would thus depend on how the lattices are used in the latent space. For the low-to-moderate dimension Euclidean sources (e.g., iid scalar sequences, Physics, and Speech), a single lattice of dimension up to 24 is used; the complexity is therefore the fixed number of operations to lattice quantize that single lattice. For images, a product lattice is applied along the channel dimension of the latent tensor, which has shape C x H x W. The number of operations would be (C/n)*H*W*(# operations for a lattice quantizer of dimension n), as there are (C/n) lattices applied H*W times, where n is the lattice dimension. Thus, for images, the complexity depends on the model architecture and image size, as C is fixed, and H, W depend on the downsampling ratio of the model and input image dimensions; the complexity again does not depend on the rate the model is operating at, unlike VQ-based methods. We have updated the paper with more details explaining this. For actual runtimes on Cheng2020 across different lattices, please refer to Table 2 in Appendix B, where we show that using the $E_8$ lattice incurs a 1.23x increase in runtime over NTC, whereas the Leech lattice incurs a 2.68x increase.
> >
> > - For LTC, generally speaking, the primary choice to be made is which lattice to use. Higher-dimensional lattices yield better performance, but have marginally higher complexity (for those used in the paper). Shown in Tables 1 and 2 in Appendix B, on Kodak, Cheng2020 with $E_8$ lattice incurs 1.23x increase in runtime over NTC but yields a -5.274\% BD-rate improvement over NTC; for Leech, the runtime increases 2.68x but yields a -16.708\% BD-rate improvement over NTC.

---

> > > ### Author Response · Authors · 2024-11-21
> > >
> > > **Q2**. In Appendix B, we have visualized quantizer cells for the 2-d Gaussian and Banana sources, evaluated on samples outside the support of the training distribution. These are provided in Figs. 13 and 14. Overall, the quantization cells successfully generalize outside the support of the training distribution, meaning that if a sample $x$ to be compressed is perturbed slightly outside the support of $p_x$, its reconstruction will still be close to $x$.
> > >
> > > **Q3**. Regarding the effect of the transforms, and interplay with the lattice quantizer, this was addressed in the ablation study of Sec. 5.1 and Appendix B.1. To clarify, the role of the transforms is to warp the quantizer tessellation used in the latent space to the source manifold. For example, in NTC, the integer lattice gets warped onto the Banana in source space, and in LTC, the hexagonal regions are warped instead. The R-D performance thus improves as a result of (i) the transforms aligning the quantizer with the source statistics and (ii) the lattice quantizer used, which combined with the transforms, yields a tessellation in source space that has superior packing efficiency. Without the transforms (Appendix B.1), lattice quantization (ECLQ) has good space-packing efficiency, but the overall compressor has worse performance as the quantizer is not aligned with the source statistics. This gap is especially large at lower rates. In addition, in Appendix B, we have added Fig. 12, demonstrating that by taking a trained Cheng2020-NTC model, and applying the Leech lattice, we already get a performance gain over Cheng2020-NTC. While using a Leech lattice quantizer with the trained NTC model improves the performance over NTC by a BD-rate of -12.984\%, training the model with the lattice quantizer helps boost the performance further to a BD-rate of -16.708\%. This shows that independent of quantization, the transforms and entropy model utilize a similar latent space; to achieve best performance, they need to be further tuned to the lattice quantizer used.

---

> > > > ### Comment · Reviewer_WNyg · 2024-11-23
> > > >
> > > > Thank you for addressing my concerns. This is a good paper.

---

### Official Review · Reviewer_mQL8 · 2024-11-03

**Soundness:** 3
**Presentation:** 3
**Contribution:** 2
**Rating:** 6
**Confidence:** 4

**Summary:**

1. This submission extends scalar-quantized non-linear transform coding of signals (Ballé et al., 2020) using lattice quantization and coding. A classical result from signal compression is that vector quantization can provide packing gains over scalar quantization especially when applied directly on correlated (i.e. non-decorrelated) signals, at the cost of higher encoding complexity. The authors propose to overcome the complexity of vector quantization using lattice quantization, which can be considered a constrained form of vector quantization. Experiments (on images and speech) were conducted to measure the gain from lattice quantization over the scalar quantization baseline.

**Strengths:**

1. The submission's attempt at marrying nonlinear transform coding (NTC) with lattice quantization (LQ) as a way of sidestepping computationally expensive vector quantization is highly relevant to the learned compression research community.

**Weaknesses:**

1. The fact that integer quantization of latent variables of NTC does not induce hexagonal quantization cells in the source domain—this is not a surprising or unexpected result, as non-linear transforms still define continuous / diffeomorphic mappings between the two domains. A transform that maps squares (scalar quantization cells in the latent domain) to hexagons (vector quantization cells in the source domain) would be highly discontinuous around cell boundaries, not a mapping that is readily learned by a neural transform in the first place.
2. Eq. (1) is central to this work, yet, no description of the Lagrange multiplier / rate–distortion tradeoff $\lambda$ appears in the manuscript. Wouldn't the choice of $\lambda$ have an impact on the characteristics/performance of the learned non-linear transform? Was an effort made to reparameterize $\lambda$ as in the work of Ballé et al. (2020)? Discussion around this is lacking in the current version of the manuscript.
3. LTC (of latent variables) inducing near-hexagonal quantization cells in the source domain (in the 2D case) seems to be due more to the diffeomorphic / continuous nature of the learned non-linear transform itself. This would mean that one could take the non-linear transform of Ballé et al. (2020) and lattice-quantize the latent variables post-hoc, and achieve similar results to what this work is proposing. The benefits of incorporating the actual lattice structure in the rate term of the optimization should be discussed.
4. Experimental results fail to clearly demonstrate the advantage of LQ over SQ. Could this be an optimization issue? For example, was $\lambda$ was fixed to some value to hurt the performance of the non-linear transform itself?

-------
4. The current presentation seems somewhat unbalanced, with the authors' actual work only starting on page 6. While this is not inherently a weakness per se, this does mean that the authors were perhaps not able to fully explore / describe the actual contributions of their work in the remaining 4–5 pages.

**Questions:**

1. Clarification surrounding the choice of hyperparameter $\lambda$ discussed in "Weaknesses" would be appreciated.
2. Experimental results: LTC performs worse than NTC at extremely low bit rates. This is surprising since the benefits of vector/lattice quantization should be more pronounced at such rates.

---

> ### Author Response · Authors · 2024-11-21
>
> Thank you for taking the time to read the paper and provide valuable feedback! Please find our responses to your comments in the following point-by-point response.
>
> **Response to weaknesses**:
>
> **W1**. We believe the value of this observation should be taken in context of the broader question of whether NTC can recover VQ (i.e., optimal) performance. This is the original question our paper seeks to answer, and is both central and important to the neural compression and information theory communities. The answer to this question is rather nuanced, as NTC consists of many components (transforms, quantizer, and learned entropy model), not just the transforms, each of which could be a contributing or limiting factor to achieving optimality. It may also depend on the source to compress. As mentioned in the introduction, prior work [2, 3] has identified certain cases in which NTC can indeed recover VQ; namely, when the source lies on a 1-d manifold. There, no component of NTC restrains it from achieving optimality. Our work examines this question for general sources, when the source lies on higher-dimensional manifolds, as is the case of iid scalar sequences. In such cases, the fact that NTC cannot recover VQ is likely due to the highly discontinuous mapping between different tessellations the transforms would need to recover, as you mentioned. The value or surprise of this observation by itself is debatable and subjective; however, in the context of our original question, this observation is more significant, as it identifies the exact reason and component determining why NTC cannot recover VQ in the more general settings we consider. A priori, we do not believe it is clear (a) if all NTC components can work together to result in optimality, or (b) if there are one or more components preventing optimality, and if so, whether they can be identified. Furthermore, we use this finding to propose the LTC framework, which successfully achieves VQ regions as an end-to-end system.
>
> **W2**. Thank you for bringing up this point. We agree that the discussion of $\lambda$ is sparse in the original manuscript, and have updated the paper accordingly around Eq. (1) and in the experimental setup. To clarify, $\lambda$ is set to a fixed value during training, which results in a learned compressor operating at a single R-D trade-off point. To get the whole trade-off for a model (all R-D plots are done this way), $\lambda$ is swept across a variety of values, with each resulting compressor’s R-D performance plotted as a point. This is the same way any variable-rate compressor is optimized, whether it is ECVQ, NTC, or LTC, and is the same way it is done in [1]. A larger $\lambda$ encourages the transforms to use more codebook vectors in the latent space (and therefore more rate). This is shown, for example, in Figs. 19-21. As an example, on iid scalar sequences, the $\lambda$ values are swept over 0.5, 1, 1.5, 2, 4, 8, resulting in the markers shown on the R-D plots.
>
> **W3**. Thanks for bringing this point to our attention. In Appendix B, we have compared a LTC model with a trained NTC model whose scalar quantizer is replaced by a lattice quantizer at inference time. The performance of this scheme is largely dependent on the source at hand (as one might expect). Fig. 11 shows that on the Gaussian source, this scheme performs the same as training LTC from scratch. This is perhaps not unexpected, as the transforms learned by NTC and LTC for the i.i.d. Gaussian source effectively perform a scaling of the source, and so interchanging them has no effect. We use the same Flow entropy model for fair comparison. In general, however, this is not expected to be the case, as the analysis transform determines which region of the lattice gets used; for more complex sources, there may be more sophisticated ways the transform determines how the lattice is used. For example, in Fig. 12, we demonstrate the same scheme on Kodak images with the Cheng2020-LTC-$\Lambda_{24}$ model. Here, we see that while using a Leech lattice quantizer with the trained NTC model improves the performance over NTC by a BD-rate of -12.984\%, training the model with the lattice quantizer helps boost the performance further to a BD-rate of -16.708\%. This result implies that one can avoid training from scratch when a good set of transforms and entropy model are already trained from a NTC architecture, and instead potentially fine-tune the NTC transforms and entropy model with the chosen lattice for best performance. In the future, when better transforms and entropy models are developed, one can also train with the lattice from scratch to get the best performance, as the training times between NTC and LTC (from scratch) are on the same scale (weeks), so there is not a significant difference in training time.

---

> > ### Author Response · Authors · 2024-11-21
> >
> > **W4**. In most sources, across most rate regimes of interest, LTC outperforms NTC rather significantly. For example, on iid scalar sequences, LTC outperforms NTC at all rates, and improves around 1dB compared to NTC for rates above 0.75. On Kodak images, LTC again outperforms NTC at all rates, and achieves up to a -16.7 BD-rate advantage compared to NTC, which measures the whole R-D tradeoff. Please refer to the new Table 1 in Appendix B, which reports all the BD-rates of all the methods compared for Kodak, and demonstrates a significant performance benefit of LTC with various lattices over NTC. Generally speaking, different rate regimes may require a different number of latent dimensions to be used for NTC and LTC. If the number of latent dimensions used by the transforms does not match the lattice quantizer dimension, this can result in suboptimal performance as the tessellation in source space may not be optimal. This is demonstrated for the Speech and Physics results (Figure 7), where different latent space dimensions (with corresponding lattices) result in different rate regimes where LTC performance “peaks”. If one takes the convex hull of all the R-D points, it results in a curve that outperforms NTC uniformly over all rate regimes. Since the integer lattice of NTC can be seen as a product lattice of a 1-d lattice, NTC performance will not differ as long as the latent dimension is larger than the latent dimensions used by the transforms. For iid sequences and images, the transforms effectively use all of the latent dimensions at all rate regimes used, so the LTC models there do not need to align the lattice dimension with rate regime for best performance. We have updated the manuscript with these discussions. Regarding $\lambda$, as mentioned in the response to weakness 2, $\lambda$ is a constant which is necessary for a particular architecture to trade off rate and distortion, so any suboptimality is not likely to be due to $\lambda$.
> >
> > **W5**. Our contributions start on page 4, where we observe NTC’s inability to recover VQ and provide explanations as to why. As mentioned in the response to weakness 1, we believe that providing an answer to “can NTC recover VQ” and pinpointing the reason behind it is a valuable contribution to the neural compression and information theory communities. In addition, we believe it is important to draw connections between this observation and other relevant fields, such as results in the companding literature that imply the performance of NTC would be limited by the choice of quantizer. These connections motivate our proposed framework in Sec. 4 in a more principled manner, rather than an ad-hoc approach to improving neural compression performance. We do not agree that we were unable to describe the contributions of the work, and have left certain experimental results less central to the work’s central questions to the appendix, such as certain ablation studies.
> >
> > Questions:
> >
> > Q1. Regarding $\lambda$, please see the response to weakness 2.
> >
> > Q2. Regarding the performance of LTC at different rate regimes, please see the response to weakness 4.
> >
> > References:
> >
> > [1] Johannes Ballé, Philip A. Chou, David Minnen, Saurabh Singh, Nick Johnston, Eirikur Agustsson, Sung Jin Hwang, and George Toderici. Nonlinear transform coding. IEEE Journal of Selected Topics in Signal Processing, 15(2):339–353, 2021. doi: 10.1109/JSTSP.2020.3034501.
> >
> > [2] Aaron B. Wagner and Johannes Ballé. Neural networks optimally compress the sawbridge. In 2021 Data Compression Conference (DCC), pp. 143–152. IEEE, 2021.
> >
> > [3] Sourbh Bhadane, Aaron B. Wagner, and Johannes Ballé. Do neural networks compress manifolds optimally? In 2022 IEEE Information Theory Workshop (ITW), pp. 582–587, 2022.

---

> > > ### Comment · Reviewer_mQL8 · 2024-11-26
> > >
> > > Thank you very much for your response, I have revised my scores and recommendation accordingly.

---

> > > > ### Author Response · Authors · 2024-11-28
> > > >
> > > > Thank you! We appreciate your engagement during the discussion period, and for the constructive feedback that has improved the paper.

---

### Official Review · Reviewer_YbH3 · 2024-11-04

**Soundness:** 3
**Presentation:** 3
**Contribution:** 3
**Rating:** 8
**Confidence:** 4

**Summary:**

This paper covers the application of lattice vector quantization to neural lossy compression.

The problem of how to address the suboptimality of scalar quantization for lossy compression has been studied for decades, and even today most practical methods use scalar quantization, and instead rely on other techniques to improve compression.

One important difference is that conventional methods normally rely on linear transform, nonadaptive methods, while this paper novelty is that it studies neural compression that uses nonlinear learned transformations. Numerical experiments demonstrate better compression results.

**Strengths:**

The authors correctly provide good evidence that the suboptimality of scalar quantization remains even with learned neural transformations.
Showing it on toy problems helps to make it clearer, but is not good at indicating how much worse on practical cases.

The paper presents examples of application to other data, and a series of experimental tests are used to answer those question on those data sources.

**Weaknesses:**

The experiments are well planned with good comparisons, but somewhat limited. Maybe too much space is used to cover the synthetic cases, and could be better used with real practical data.

It would also be better to have more detailed analysis of the results in those cases. Comments like “…with the Leech significantly outperforming VTM and NVTC,” are too vague and not accurate.

It looks like there are good gains with the modifications to the Cheng2020 method, which may be the most important conclusion of the paper, and that should be more carefully reported.

**Questions:**

It would be interesting to see more information about the practical aspects of using lattice quantizers during training and for actual coding.

---

> ### Author Response · Authors · 2024-11-21
>
> Thank you for taking the time to read the paper and provide valuable feedback! Please find our responses to your comments in the following point-by-point response.
>
> **Response to weaknesses**:
>
> **W1**. We acknowledge that synthetic sources are a significant part of our experiments. We do work with real-world data, ranging from scientific and audio to images, with significant gains achieved for image compression. The primary goal of our work is to understand the fundamental limitations of NTC, and how to improve the performance to approach optimality. If optimality is achieved, then there is no more room for improvement. Such a fundamental analysis requires synthetic sources where optimality is known, and all causes for potential gains or suboptimality can be isolated. While real-world data may be more realistic, there are many factors that may contribute to differences in R-D performance, which precludes a fundamental analysis. Our set of experiments demonstrate the suboptimality of NTC (requiring synthetic data), ability of LTC to approach the optimal (requiring synthetic data), and the superiority of LTC over NTC (shown on both synthetic and real-world data). In addition, our experiments on real-world data do cover a broad range of data modalities, ranging from scientific and audio data to images, demonstrating the broad applicability of LTC to improve upon NTC in practical scenarios.
>
> **W2**. Thank you for your suggestion. We have added a BD-rate table in Appendix B, with both VTM and Cheng2020-NTC as an anchor, and additionally revised the text to provide precise comparisons between methods. To summarize, (i) while NVTC achieves a -2.919 BD-rate gain over VTM, Cheng2020 with Leech is significantly better, with a -11.895\% BD-rate gain, and (ii) the increasing BD-rate gains of Cheng2020-LTC with lattice dimension further supports evidence that the rate-distortion performance is significantly determined by the quantization choice in the latent space, and that the NTC model is able to leverage the enhanced packing efficiency of higher-dimensional lattices.
>
> **W3**. This is now addressed by the more precise comparisons provided in the previous point. We agree that good gains achieved with Cheng2020 are an important conclusion. We hope that the BD-rate table and discussions help highlight these findings in addition to the fundamental analysis provided in this paper, such as understanding of where and when NTC methods are suboptimal, and why and how LTC resolves them. Those conclusions are also valuable because NTC/LTC can be applied to many modalities (not just images), so such an understanding impacts a broader range of research than just image compression.
>
> **Response to questions**:
>
> **Q1**. Thank you for your suggestion. We added some more details on the practical aspects of lattice quantizers in the revised manuscript, in the implementation details in the appendix. For the low-to-moderate dimension Euclidean sources (e.g., iid scalar sequences, Physics, and Speech), a single lattice of dimension up to 24 is used. Thus a single call to the lattice quantizer’s CVP algorithm is required to compute the quantized latent, as well as a CVP call for each Monte Carlo sample for the entropy model. For images, a product lattice is applied along the channel dimension of the latent tensor, which has shape C x H x W. There are (C/n) lattices applied H x W times, where n is the lattice dimension. We do not evaluate entropy coding in this paper as it is not the focus of our paper, and we leave this to future work; as mentioned in Sec. 5, the rates reported are the cross-entropies of the learned entropy model. However, we do not expect this to increase complexity significantly for the lattices used in this work. For example, for the E_8 lattice used for image compression (Fig. 7), the effective codebook size will be around 2^(8*bpp), which is a codebook size commonly seen in reduced-complexity VQ methods such as NVTC [1], and there are also 8x less symbols to code compared to scalar quantization. Additionally, Sec. 4.2 discusses the use of nested LQ, which provides a low-complexity solution to encoding with lattices.
>
> References:
>
> [1] Runsen Feng, Zongyu Guo, Weiping Li, and Zhibo Chen. Nvtc: Nonlinear vector transform coding.
> In Proceedings of the IEEE/CVF Conference on Computer Vision and Pattern Recognition, pp.
> 6101–6110, 2023.

---

> > ### Comment · Reviewer_YbH3 · 2024-11-26
> >
> > Thank you for considering my comments and questions and preparing the rebuttal. Your responses address all my main concerns, and thus I am raising my score.

---

> > > ### Author Response · Authors · 2024-11-28
> > >
> > > Thank you! We appreciate your engagement during the discussion period, and for the constructive feedback that has improved the paper.

---

### Official Review · Reviewer_quJZ · 2024-11-04

**Soundness:** 3
**Presentation:** 3
**Contribution:** 3
**Rating:** 8
**Confidence:** 4

**Summary:**

This paper proposes lattice transform coding (LTC), which uses lattice quantization beyond the conventional scalar quantization in neural lossy compression. The paper offers evidence that learned transforms in conventional nonlinear transform coding (NTC) are insufficient in overcoming sub-optimal quantization in the latent space, and shows that LTC with more efficient space-filling lattices indeed bring improvements in R-D performance. Besides new training and rate estimation techniques for LTC, the paper also proposes nested LTC to lower the practical computation cost of entropy coding, as well as block LTC to perform block coding in order to close the gap to the information-theoretic optimal performance (R(D)).
Experiments on various synthetic and real-world data sources support the effectiveness of the proposed LTC approach and give insight to the suboptimality of current NTC approaches.

**Strengths:**

I consider the originality of the paper to be moderately high. Although the ideas of lattice coding, nested quantization, block coding, etc. have been known for a while, and lattice coding itself even explored in the contemporary work of Kudo et al. (2023), this paper brings these ideas together and gives novel insights for the performance gap between the current NTC methods and the theoretical optimum of R(D).

The paper is well-written and mostly easy to follow (although see issues below), with insightful explanations and experiments.

I consider the contribution of this paper timely and significant. Unlike much of current research on NTC which is in architectural / entropy modeling improvements, this paper examines a fundamentally different component of NTC and makes a compelling case of the advantage of lattice quantization over scalar quantization.  Achieving the theoretical advantage of block coding / VQ has been a non-trivial goal of lossy compression, and this paper offers valuable perspectives and a potential path forward.

**Weaknesses:**

1. A significant difficulty of lattice coding seems to be the complexity of implementing entropy coding. Nested lattice quantization was proposed but appears to come with a significant penalty to R-D performance. Therefore most of the R-D curves for LTC in this paper are theoretical. I don't expect this problem to be solved in this paper, but I think it'll be useful to give insights for how to operationalize the reported R-D curves, in particular considerations for deriving the discrete entropy model from the normalizing flow prior in a way that is reproducible by the decoder.

2. Some of the the explanations around nested LQ and block LQ were a bit hard to follow. For example, the first two sentences on lines 334 - 335 seemed to come out of nowhere ( "The goal is for the majority of the latent mass to be placed in the Voronoi region of Λc. This happens when QΛc (yf ) = 0."). Similarly, line 375 introduces a new pair of "analysis/synthesis transforms of LTC", denoted $c_a^{(i)}, c_s^{(i)}$ --- why are they necessary, given that we have already operate in the transformed space using $g_a, g_s$? And how should we choose  $c_a^{(i)}, c_s^{(i)}$ if they are indeed necessary?

Also, a minor typographical error: there seems to be inconsistency between "n" and "d": line 368 explains "Note that we we retain n as the block-length parameter, but use d as the dimension of x." So should line 458 write "d = 16 and 32"?

**Questions:**

1. I'm confused by how LQ is applied when the source dimension does not equal the lattice dimension. It seems the lattice dimension used in this paper only goes up to 24, but the source dimension can be 33 (Speech) or even higher (Kodak). Is it based on some kind of product lattice then?


2. Related to above, in the block LTC scheme (Sec 4.3), what is the motivation for grouping columns of the transform coefficients matrix into length-n blocks? Given that the data $x$ and the resulting latent tensor $y$ is often high dimensional, can we do block coding on length-n blocks of $y$ in some other ways? I would expect more gains if the entries in each block are **not** i.i.d.

3. Any idea why the R-D performance is sometimes worse when a bigger latent space ($d_y$) is used, all else being equal (e.g., the $d_y=4$ curve is worse than $d_y=2$ at low rates in the Physics Figure 7)? The usual observation in NTC is that the R-D performance can be improved by increasing the latent dimensionality when the latter is too small (esp. towards the low distortion regime), but further increasing the latent dimensionality beyond a sufficient capacity generally does not harm the R-D performance.

---

> ### Author Response · Authors · 2024-11-21
>
> Thank you for taking the time to read the paper and provide valuable feedback! Please find our responses to your comments in the following point-by-point response.
>
> **Response to weaknesses**:
>
> **W1**. To operationalize the R-D curves, the interpretation would resemble that for NTC methods (e.g., see Fig. 10 in [1]). That is, one defines a discrete PMF over the quantized codebook vectors via integrating a parameterized continuous entropy model; for LTC, this is defined in Eq. 5. The continuous entropy model would be available at both the encoder and decoder to provide likelihoods for entropy encoding and decoding. To ensure reproducibility of the likelihood values in Eq. 5 at both the encoder and decoder, a shared random seed could be either predetermined or transmitted as part of the bitstream.
>
> **W2**. Thanks for the feedback, and we have clarified the writing of nested LQ (Sec. 4.2) and block LTC (Sec 4.3) in the text. We hope the following comments help resolve your questions. Regarding nested LQ, the high-level idea is to use the coarse lattice to define a bounded region of the fine lattice. The codebook vectors of the fine lattice, contained in the said region, are the ones we wish to use to encode the source via index coding. This happens when the majority of the latent mass is placed in the coarse lattice cell centered at the origin. Regarding block LTC, the idea is for $g_a$ and $g_s$ (which are shared across the block) to map each source vector to a lower-dimensional latent space of $d_y$ dimensions, which is often sufficient; for example, the Sawbridge is mapped from 1024 dimensions to 1 or 2 dimensions. We now have a sequence of latent i.i.d. vectors. If we directly apply LQ across the i.i.d. slices, this would not result in an optimal compression, as it is equivalent to applying LQ directly to i.i.d. sequences, which is suboptimal (mentioned in the ablation study, Sections 5.3 and B.1) for any source that does not have a uniform density. The $c_a$, $c_s$ transforms help shape the lattice regions to match the statistics of the iid latent source; for the i.i.d. vector sources in this paper, a small MLP is sufficient, similar to the i.i.d. scalar sequences in Sec. 5.1.
>
> Thanks for pointing out the typo. Line 458 should indeed say “d=16 and 33”. We have updated the paper accordingly.

---

> ### Author Response · Authors · 2024-11-21
>
> **Response to questions**:
>
> **Q1**. When the source dimension does not equal that of the source, there are several options, depending on the model architecture and source dimensions. For Euclidean vector sources, such as Speech and Physics, we map the source to a latent dimension of $d_y$, and use a single lattice of dimension $d_y$. If $d_y$ is greater than 24, which we do not use in this paper for Speech and Physics, then a product lattice could be used. For images, since the source is now a tensor, the latent space is also a tensor of shape C x H x W, where C is the channel dimension, and H, W are the spatial dimensions. As mentioned in the paper, we apply LQ to the channel dimension. C ranges from 128 to 320, depending on the model architecture; we thus use product lattices for E8 and Leech. The $D_n^*$ lattice is defined for any dimension, so it does not require a product lattice. We have clarified this in the paper in the experimental details.
>
> **Q2**. The goal of block LTC is to design the lowest complexity architecture that will capture gains from compressing iid blocks simultaneously. There are two options: (i) block LTC as it is currently defined, and (ii) a block LTC method that jointly maps the iid vector sequence to a larger latent space. For (i), since the source is a iid sequence of vectors, they can all share a common latent space, which only requires g_an and g_s to be applied sample-wise. Now we have a iid sequence of latent vectors $y_i$. If we apply LQ with each $y_i$ vector, this will be no different from one-shot coding, so that is why LQ is applied across the $n$-length blocks (see response to weakness 2). The other option (ii) would be to not use a sample-wise $g_a$ and $g_s$, and instead use transforms that jointly map the iid block to a much larger latent space of $n*d_y$. Then, the iid structure is likely lost and we could perhaps get more gains by applying LQ in other ways. However, such a $g_a$ and $g_s$ would not scale well with $d$ or $n$ as they are implemented with MLPs for the sources we use block LTC for (i.e., parameter count grows $O((dn)^2)$). With idea (i), the g_a transforms have $O(d^2)$ parameters, and the $c_a$, $c_s$ transforms have $O(n^2)$ parameters. Note that option (ii) is quite similar conceptually to how the image compression models work, except convolutional layers are used to have a more efficient parameter count, the image’s pixels (which can be thought of as a sequence of vectors) are not iid, and the lattice is indeed applied across vectors that should be correlated in the latent space. Future work could explore option (ii) to achieve further gains in block coding. We hope this discussion clarifies your question.
>
> **Q3**. Generally speaking, different rate regimes may require a different number of latent dimensions to be used for NTC and LTC. If the number of latent dimensions used by the transforms does not match the lattice quantizer dimension, this can result in suboptimal performance as the tessellation in source space may not be optimal. This is demonstrated for the Speech and Physics results, where different latent space dimensions (with corresponding lattices) result in different rate regimes where LTC performance “peaks”. If one takes the convex hull of all the R-D points, it results in a curve that outperforms NTC uniformly over all rate regimes. Since the integer lattice of NTC can be seen as a product lattice of a 1-d lattice, NTC performance will not differ as long as the latent dimension is larger than the latent dimensions used by the transforms. For iid sequences and images, the transforms effectively use all of the latent dimensions at all rate regimes used, so the LTC models there do not need to align the lattice dimension with rate regime for best performance. We have updated the revised manuscript in Sec. 5.1 with these discussions.
>
> References:
>
> [1] Johannes Ballé, Philip A. Chou, David Minnen, Saurabh Singh, Nick Johnston, Eirikur Agustsson, Sung Jin Hwang, and George Toderici. Nonlinear transform coding. IEEE Journal of Selected Topics in Signal Processing, 15(2):339–353, 2021. doi: 10.1109/JSTSP.2020.3034501.

---

### Author Response · Authors · 2024-11-21
**Initial common response to reviewers**

We appreciate the reviewers taking the time to read our paper and provide helpful comments and feedback. We thank the reviewers for recognizing our work as one that provides valuable insights on the role of quantization and fundamental performance gap to optimality, highly relevant to the learned compression community, timely, and significant. To address the reviewers' concerns, we have responded to each reviewer's comments below individually with a point-by-point reply, and hope they help alleviate the reviewer's concerns. The manuscript has also been updated to reflect suggested changes, with new text edits shown in blue. New experiments, tables, and figures addressing questions or concerns have been added to Appendix B.

---

### Meta-Review · Area_Chair_JVdD · 2024-12-16

**Metareview:**

The authors discuss a lossy compression method based on transform coding. In contrast to the rectangular lattices used in the typical nonlinear transform coding (NTC) setup, the authors propose to use higher dimensional lattices and demonstrate that even in the context of nonlinear transforms, significant gains can be achieved.

The main limitation of the proposed method is in the practical implementation, which is less straight-forward compared to NTC, as noted by the reviewers (e.g., complexity of the entropy model, determining probabilities of each quantization bin in an efficient way). The authors should address these limitations in the final version of the manuscript.

**Additional Comments On Reviewer Discussion:**

The review was fairly straight-forward. Reviewers noted that this is a good paper, and mainly asked for clarification on certain points, which the authors addressed. The main limitation in my view is the practical implementability, which was also noted by the reviewers, and remained not satisfactorily addressed.

---

### Decision · Program_Chairs · 2025-01-22

Accept (Spotlight)